# Calving cycle of the Brunt Ice Shelf, Antarctica, driven by changes in ice-shelf geometry

Jan De Rydt[1], G. Hilmar Gudmundsson[1], Thomas Nagler[2], Jan Wuite[2]

[1]Department of Geography and Environmental Sciences, Northumbria University, Newcastle upon Tyne, UK.
[2]ENVEO – Environmental Earth Observation, Innsbruck, Austria.

*Correspondence to*: Jan De Rydt (jan.rydt@northumbria.ac.uk)

**Abstract.** Despite the potentially detrimental impact of large-scale calving events on the geometry and ice flow of the Antarctic Ice Sheet, little is known about the processes that drive rift formation prior to calving, or what controls the timing of these events. The Brunt Ice Shelf in East Antarctica presents a rare natural laboratory to study these processes, following the recent
formation of two rifts, each now exceeding 50 km in length. Here we use two decades of in-situ and remote sensing observations, together with numerical modelling, to reveal how slow changes in ice shelf geometry over time caused build-up of mechanical tension far upstream of the ice front, and culminated in rift formation and a significant speed-up of the ice shelf. These internal feedbacks, whereby ice shelves generate the very conditions that lead to their own (partial) disintegration, are currently missing from ice flow models, which severely limits their ability to accurately predict future sea level rise.

## 1 Introduction

Icebergs that calve from the floating margins of the Antarctic Ice Sheet account for up to 50% of ice discharge into the Southern Ocean (Depoorter et al., 2013). The largest calving events, such as the loss of a 5000 km$^2$ iceberg from the Larsen C Ice Shelf in 2017 (Hogg and Gudmundsson, 2017), result from the horizontal lengthening of multi-kilometre long rifts that cut through the full thickness of the ice. These large-scale events, in contrast to the loss of small ice blocks in the bending zone near the
ice front (Reeh, 1968), significantly reshape the geometry of the ice-shelf margins, and can have a profound impact on their structural integrity (Doake et al., 1998). Because ice shelves act as a barrier around the grounded ice and buttress its seaward flow through lateral drag and local grounding in shallow water (Dupont and Alley, 2005), any loss of buttressing around the periphery of Antarctica as a result of calving-induced changes in ice-shelf geometry will adversely affect glacier flow (Scambos et al., 2004, Rignot et al., 2004, Rott et al., 2011), and induce additional ice discharge into the Southern Ocean.

Larger calving events are part of the natural life cycle of all ice shelves, as they go through internally-driven periods of growth and collapse (see e.g. Fricker et al., 2002, Anderson et al., 2014, Hogg et al., 2017). Despite the importance of calving for the mass balance of the Antarctic Ice Sheet, detailed observations of such events and related changes in ice-shelf dynamics remain scarce. In particular, conditions for full-depth fracture and the subsequent propagation of kilometre-scale rifts are poorly

understood. Previous studies have suggested that glaciological stresses are a major control on rift formation and propagation, see e.g. Fricker et al., 2002, Joughin et al., 2005, Larour et al. 2004, Borstad et al., 2012, 2017, and the build-up of internal stresses within an ice shelf generate energetically favourable conditions for the formation and propagation of rifts that cut through the full depth of the ice column (Rist, 2002). However, a direct link between changing stress conditions prior to

calving, and the location and timing of rifts has not been demonstrated so far. This is in part due to the long characteristic time scales over which stresses evolve (typically multiple decades), and the lack of observational data required to calculate the stresses over the duration of a full calving cycle.

Once initiated, ice shelf rifts have been shown to lengthen at rates that vary strongly in time (Bassis et al., 2005, Walker et al.,

2015, Borstad et al., 2017, De Rydt et al., 2018), from meters to kilometres per day, and can arrest for extended periods. Suture zones of basally accreted marine ice have been linked to periods of slow rift propagation and could delay or halt large-scale calving (Borstad et al., 2017). Contrasting observations have reported fast unzipping of rifts along bands of marine ice and slow propagation through meteoric ice (De Rydt et al., 2018, King et al., 2018), highlighting the complex nature of rift behaviour. At present, a unified formulation of rift dynamics rooted in existing theory of fracture mechanics is still under

development (Rist et al., 2002, Bassis et al., 2008, Lipovsky, 2018). As a result, predictions for the location and timing of large-scale calving events remain ill-constrained and the feedback between calving rates and ongoing climate-change induced thinning of ice shelves (Pritchard et al., 2009, Flament and Remy, 2012, Konrad et al., 2016) or changes in the internal structure of the ice are unknown.

Fortuitously, a new opportunity to enhance our process-based understanding of rift dynamics and calving has recently arisen with the impending calving of two tabular icebergs from the Brunt Ice Shelf (BIS) in East Antarctica (Figure 1a). In December 2012 a historical rift structure, 'Chasm 1', that had lain dormant for three decades (Figure 1b), was reactivated and started to lengthen by several kilometres per year (De Rydt et al., 2018). The renewed rifting activity was followed by the formation of a second rift, the so-called 'Halloween Crack' in October 2016 (De Rydt et al., 2018), which grew quickly and reached a length

of 60 km by October 2018 (Figure 1b). Both Chasm 1 and the Halloween Crack continue to grow to date.

Changes in dynamics of the BIS have been documented in great detail before and after rift formation, and the behaviour of both rifts has been monitored closely since the day of their initiation, in part by an extensive network of up to 15 permanent GPS stations (Gudmundsson et al., 2017, De Rydt et al., 2018). Furthermore, recent advances in satellite data availability have

provided a comprehensive spatial and temporal description of the flow and ice deformation across the ice shelf (Thomas, 1973, Simmons and Rouse, 1984, Simmons, 1986, Gudmundsson et al., 2017). In addition, the continuous occupation of the Halley Research Station on the ice shelf since the mid-1950s and a long-term glaciological monitoring programme, has allowed us to put ongoing changes into a historical perspective.

The long-term observational record of the BIS provides unprecedented coverage of glaciological changes over a full calving cycle, from the last calving event in the early 1970s to present day. Based on this record and earlier observations of the ice front location in 1915, 1958 and 1986 (Anderson et al., 2014) and flow speed measurements since the 1950s (Gudmundsson et al, 2017), a cyclic pattern of glaciological changes emerged. First, calving and ice front retreat caused a loss of pinning from a small seabed shoal (McDonald Ice Rumples or MIR in Figure 1a), which triggered an acceleration of the flow. Subsequently, expansion of the ice shelf and local regrounding at the MIR lead to a slow increase in buttressing and deceleration of the flow. As such, the cyclic dynamics of the BIS is modulated by natural changes in ice shelf geometry, and observations indicate that each cycle lasts approximately 40-50 years, which is comparable to other stable ice shelves (see e.g. Fricker et al., 2002).

The significance of local grounding at the MIR for the dynamics of the BIS, and its role in recent rifting events will be explored in subsequent chapters. However, the wider importance of pinning points for the dynamics and structural integrity of Antarctic ice shelves has been previously recognised (Borstad et al., 2013, Matsuoka et al., 2015, Favier et al., 2016, Berger et al., 2016, Gudmundsson et al., 2017), and their potential role in triggering calving events was highlighted recently for Pine Island Glacier (Arndt et al, 2018). Here we use the BIS as an example to demonstrate the link between naturally evolving glaciological conditions, the initiation of ice-shelf rifts, and the mechanical drivers that govern subsequent rift propagation. The geometrical configuration of the BIS is not unique, and similar principles likely apply to other Antarctic ice shelves that are dynamically constrained by local pinning points, such as the ice shelves in Dronning Maud Land (Favier et al., 2016) and the Larsen C Ice Shelf (Borstad et al., 2013). Our study is more generally relevant for ice shelves that experience a build-up of stress, potentially far upstream of the ice front, due to natural changes in ice-shelf geometry.

The paper is organized as follows. In section 2 we provide a brief overview of the historical and ongoing glaciological changes on the BIS, and argue how these changes are part of the natural lifecycle of the ice shelf. In section 3 we introduce the tools and data that were used to diagnose the glaciological conditions that gave rise to the initiation and propagation of Chasm 1 and the Halloween Crack. The results are divided into 3 parts: in section 4 we present a timeline of changes in glaciological stress that led to the initiation of the rifts; in section 5 we discuss the drivers of subsequent rift propagation; in section 6 we compare the observed dynamical changes before and after rift formation to model projections, and quantify model errors related to the absence of a suitable calving law. Conclusions are presented in section 7.

**2 Historical context and the calving cycle of the Brunt Ice Shelf**

In the early 1970s, a single calving event significantly reduced the extent of the BIS (Thomas, 1973), and the retreat of the ice front caused a loss of contact between the ice base and a seabed shoal at the McDonald Ice Rumples (MIR, Figure 1a). The localized loss of friction with the seabed resulted in a reduction of the backstress (or buttressing) and coincided with a twofold increase in flow speed of the remainder of the ice shelf (Simmons and Rouse, 1984, Gudmundsson et al., 2017). This speed-

up unequivocally demonstrated the potential impact of geometrical changes on ice-shelf dynamics. In decades following, the ice front re-advanced by up to 30 km in places (Figure 1b) and the deforming ice gradually re-established ice-to-bed contact at the MIR, causing the ice shelf to slow down to pre-calving speeds by 2012 (Gudmundsson et al., 2017).

The dramatic succession of speed-up and slow-down by over 100% within a few decades comprises some of the highest amplitude variations in flow speed observed in Antarctica, and we argue that these changes are driven entirely by internal ice dynamics processes. The BIS, which is situated on the eastern edge of the Weddell Sea, has not been affected noticeably by changes in external conditions during recent decades. Sustained negative surface temperatures throughout the year prevent surface melting (Anderson et al., 2014) and eliminate the risk of crevasse penetration caused by hydrofracturing (Scambos et
al., 2000) and potential weakening of the ice shelf. There is also no indication that offshore Modified Warm Deep Water intrudes into the ice shelf cavity to cause significant basal ablation (Nicholls et al., 2009) or ice-shelf thinning (Paolo et al, 2015). As a result, the BIS represents a unique setting in which large-scale calving processes can be studied in relative isolation, and the wealth of available data can be probed to gain unbiased insights into the mechanics of ice-shelf fracture.

In 2012, following four decades of ice shelf growth, the ice front of the BIS reached its most advanced position since the beginning of measurements in 1915 (Anderson et al., 2014). The advance of the ice front coincided with local grounding of the ice shelf at the MIR over a small but prominent area of 5 km$^2$. At the same time, preconditions for rifting were re-established (as will be explained in section 4), and the reactivation of Chasm 1 in December 2012 and formation of the Halloween Crack in October 2016 marked the start of two new calving events. Their combined impact is expected to reduce the ice shelf's area
by more than 50% (De Rydt et al., 2018), the largest single perturbation in ice shelf geometry on record. In response to the damage caused, a renewed increase in flow speed by up to 10% per year was observed between 2012 and present-day across most of the ice shelf (Figure 1b and (Gudmundsson et al., 2017)).

Based on this 50-year record of ice geometry and flow speed, the BIS appears to exhibit successive phases of fast acceleration
and slow deceleration of the flow, modulated by changes in geometry and buttressing at the MIR. In subsequent sections we investigate how these changes in glacio-mechanical conditions led to the reactivation of Chasm 1 in 2012 and caused the initiation of the Halloween Crack in October 2016 at the observed location.

## 3 Methods and data

In order to investigate the timing and location of rifting in relation to mechanical changes in the ice shelf, we use spatial maps
of principal stress magnitude and direction as a diagnostic tool. The maximum principal stress magnitude is used in fracture mechanics as a criterion to identify when brittle materials fail under tension, see e.g. Rist et al., 2002. Although we do not aim to formulate the details of such a fracture criterion here, or discuss complications due to the brittle-ductile properties of ice,

we acknowledge the potential effect of high stress concentrations (or load) on the structural integrity of the ice. We analyse maximum principal stress patterns for 9 different configurations of the BIS between 1997 and 2018, based on snapshot observations of surface velocity, ice thickness and ice shelf extent (Table 1). We subsequently relate spatiotemporal changes in the principal stress to the observed timing and location of rifting events.

## 3.1 Calculation of horizontal stresses

The elements of the stress tensor cannot be measured directly, but are related to the strain rates $\dot{\epsilon}$ and rate factor $A$ through the material rheology, described by Glen's Law:

$$(1) \qquad \dot{\epsilon} = A\tau_E^{n-1}\tau,$$

with $\tau$ the deviatoric stress tensor and $n=3$ the creep exponent. In ice bodies with a uniform rate factor $A$, horizontal strain rates can be calculated directly from observed surface velocities, and Eq. (1) implies an estimate for the deviatoric stresses and the associated principal components.

However, in reality the rate factor $A$ varies spatially over several orders of magnitude (both horizontally and vertically), and an alternative approach for estimating $\tau$ relies on commonly-used inverse theory, which uses observations of ice shelf geometry, velocity and ice thickness data to estimate an optimal spatial distribution of $A(\vec{x})$ by minimizing the mismatch between observed and simulated ice velocities (see e.g. MacAyeal, 1993 and Larour et al., 2005). The resulting solution for $A(\vec{x})$ and the diagnostic model output for $\dot{\epsilon}$ can be used to calculate a spatial distribution of the deviatoric stress $\tau$ and its principal components. Here, we used an adjoint iterative optimization method with Tikhonov regularization within the SSA (Shallow Shelf Approximation) ice flow model Úa (Gudmundsson et al, 2012) to obtain vertically-integrated values for $A(\vec{x})$ and $\tau(\vec{x})$, where $\vec{x}$ denotes both horizontal dimensions. Further details about the model setup, the inversion procedure and examples of $A(\vec{x})$ for various ice-shelf configurations can be found in Appendix A.

## 3.2 Observational datasets

The inverse method requires input from three key observational datasets: ice thickness, surface velocity and ice-shelf extent (i.e., ice front and grounding line location). In this study, inversions for 9 successive ice shelf configurations between 1997 and 2018 were carried out, giving 9 snapshots of the horizontal stress distribution in the ice shelf. More details about the data sources for each of these configurations can be found in Table 1. Additional data from intermediate times, in particular MEaSUREs and Sentinel-1 velocity fields, are available and can be used to obtain a denser time series of stress patterns. However, analysis of the additional data did not contain any new findings beyond those presented.

*Ice thickness* estimates were derived from a digital elevation model (DEM) and a flotation criterion assuming a two-layer density model with a 30 m firn layer ($\rho_{firn}$= 750 kg/m$^3$) overlaying solid ice ($\rho_{ice}$= 920 kg/m$^3$) (De Rydt et al., 2018). For the 01/01/1999 stress calculation, the Bedmap 2 surface DEM was used. For all other stress calculations, a new DEM was generated from a mosaic of 3 m horizontal resolution WorldView-2 tiles acquired between 19 October 2012 and 30 March

2014 (covering the Brunt Ice Shelf) and Cryosat-2 data (Slater et al., 2018) (covering the Stancomb-Wills Glacier Tongue). To correct for the ice motion between the different acquisition times of the WorldView-2 tiles, all tiles were translated to a common datum of 1 January 2013. For each tile, pixels were shifted by $\Delta\vec{x} = \vec{u}\,\Delta t$ with $\vec{u}(\vec{x})$ the surface velocity at location $\vec{x}$ obtained from a pair of Sentinel-1 images acquired in June 2015, and $\Delta t$ the difference between the acquisition time of the tile and the common datum. Subsequently, a constant vertical shift was applied to each tile to minimize the misfit in

overlapping regions. The resulting surface DEM was compared to 5000 km of in situ airborne Light Detection And Ranging (LiDAR) data acquired in January 2017 (Hodgson et al., 2019) and referenced with respect to sea level using 8 LiDAR sections over the open ocean. The mean difference between the resulting DEM and LiDAR data in overlapping regions was 0.01±3.6 m.

*Surface velocity* data were acquired from a variety of sources, as detailed in column 3 in Table 1. Velocity fields based on Sentinel-1 data were processed using an iterative offset tracking method (Nagler et al., 2015). To account for tidal artefacts, all velocity maps were cross-calibrated to high-precision GPS data from a long-term network of up to 15 dual frequency GPS receivers on the BIS (Anderson et al., 2014, Gudmundsson et al., 2017, De Rydt et al., 2018 and Figure 1a). The GPS data were processed using PPP techniques using the Bernese software to obtain daily positions with sub-centimetre precision. For

each horizontal velocity component, a linear regression between satellite data and GPS displacements over the corresponding satellite acquisition period was used to calculate the mean offset between both datasets. The offset was subtracted from satellite-derived estimates of surface velocity in order to ensure an optimal fit between the latter and in situ GPS data. Each final velocity product was assigned an effective timestamp corresponding to the middle of the feature tracking window (first column in Table 1). In order to guarantee consistency between the surface velocity and DEM in the model inversion, the DEM

was translated to the velocity timestamp using the method described above.

*Ice front positions* were outlined from Landsat-7/8 cloud-free panchromatic band images (column 4 in Table 1). The temporally varying extent of grounding at the MIR was derived from a combination of proxy indicators, in particular crevasse patterns, surface velocity data and surface elevation. Due to the inaccessibility and complex topography of the surface at the MIR,

ground-based and airborne radar surveys have failed to reliably measure the bedrock topography in this location (Hodgson et al., 2019). In our analysis, the elevation of the bed was therefore set to 10 m above the floatation depth across the extent of the MIR, and basal traction between the bed and ice was parameterized by a Weertman sliding law. The latter provides a commonly adopted relation between the basal sliding velocity $v_b$ and basal shear stress $\tau_b$ in grounded areas, $\tau_b = C^{-1/m}\|v_b\|^{\frac{1}{m}-1}v_b$, with

*m* and *C* model parameters. A common value for the sliding exponent *m* = 3 was chosen, and the slipperiness coefficient was set to a spatially uniform value $C = 10^{-3}$.

## 4 Ice-shelf growth causes rifting

Figure 2 shows a time series of the principal stress directions (arrows) and maximal principal deviatoric stress (colours) in the horizontal plane, covering 12 years before to 4 years after the reactivation of Chasm 1 in December 2012. Before 2000 (Figure 2a), the stress pattern is characteristic for a nearly free floating (or unbuttressed) ice shelf, with most areas showing extensive deviatoric stresses in both principal directions. Note that at this time, grounding at the MIR was restricted to a small area of about 1 km$^2$, which caused higher-than-average stresses at the ice front in this area (Figure 2a), but limited upstream buttressing. Between 2000 and 2007 (Figure 2b), a fast and sharp transition occurred from a purely tensile to a mixed tensile/compressive regime, with compressive stress trajectories radially aligned around the MIR. This pattern is characteristic of a point pressure source located at the MIR, and supports the notion that growing contact between the ice base and sea floor caused an increase in backpressure in this area. With the onset of compression, tensile stresses increased by more than twofold, with the largest values found 10 km upstream of the MIR. Between 2007 and 2013 (Figure 2c), the zone of high tension expanded and spread outward from the MIR, with values up to 120 kPa. Once the periphery of this zone reached the historical rift tip of Chasm 1 in December 2012, the ice shelf eventually fractured along this line of pre-existing weakness (Figure 2c).

After the initiation and sub-critical propagation of Chasm 1 in December 2012, stress values on the western shelf significantly reduced between 2013 and 2016. Simultaneously, bands of high tension developed towards the south and east of the MIR (Figure 2d) with estimated tensile deviatoric stress values up to 140 kPa. These bands show no obvious spatial correlation to variations in ice thickness or internal ice structure (King et al., 2018). On 4 October 2016, the ice shelf fractured within the band nearest and about 15 km upstream of the MIR (Figure 2d). Following rift initiation, the Halloween Crack rapidly propagated towards the MIR and in the opposing eastward direction along trajectories perpendicular to the local maximal tensile stress direction (Figure 3a and b).

Our calculations provide a simple and intuitive explanation for the sudden reactivation of Chasm 1 in December 2012 and the formation of the Halloween Crack in October 2016. The timing of both rift initiations, the location and the subsequent propagation paths can all be explained in relation to the magnitude and orientation of the tensile deviatoric stress distribution (Figure 2). In both cases, the rifts formed in response to a gradual build-up of horizontal tensile stresses that took place over decades as the ice shelf expanded over time, and increased its contact with the seabed at the MIR. The locations of initiation were consistent with the hypothesis that ice-shelf areas subjected to the highest tensile stress are most susceptible to failure. A priori, these favourable conditions, dictated by changes in geometry, are not restricted to areas close to the ice front or within the shear margins. In particular, they can occur landward of the compressive arch, which is the transition zone between freely

floating (or passive) ice close to the ice front, and upstream ice in compression (Doake et al., 1998, Fürst et al., 2016). Rifts that cut through the compressive arch, as is the case for the Halloween Crack, will affect the buttressing capacity of the ice shelf, and thereby induce changes in ice shelf dynamics or continue to affect its structural integrity (Doake et al., 1998).

## 5 Discontinuous rift propagation

Following the initiation of both rifts, the sustained deformation of the ice shelf's geometry and reduction in load-bearing capacity due to rift propagation, caused progressive changes in the large-scale stress pattern. As previously noted, the formation of Chasm 1 coincided with an increase in deviatoric stress towards the south and east of the MIR, which likely contributed to the formation of the Halloween Crack (Figures 2c-d). Following rift propagation, newly formed rift surfaces were subjected to ocean pressure, and forces within the ice shelf adjusted to the new boundary conditions and newly emerging ice front
location. In particular, maximum tensile stresses aligned perpendicular to the edges of the rifts.

In Figure 3, principal stress patterns for 5 different ice shelf geometries between December 2016 and October 2017 are shown, demonstrating the changes as Chasm 1 and the Halloween Crack propagated. In November 2016, the tip of the Halloween Crack stagnated within a prominent zone of high tensile stress (Figure 3a) for a 4-week period, despite persistent rift widening
(De Rydt et al., 2018). It was previously noted that this area consists of a complex conglomerate of thick meteoric ice and thinner marine ice (De Rydt et al., 2018, King et al., 2018), and such inhomogeneities have the potential to slow down rift propagation. From around 15 December 2016, the period of slow changes in rift length and high concentrations of remotely-applied stress was followed by a period of fast propagation as the rift cut through an area of relatively homogeneous marine ice. By 29 December 2016, the Halloween Crack had propagated a further 11 kilometres at an average rate of 800 m/day
(compared to <100 m/day in November). Following this event, a significant reduction in the calculated tensile stress indicated an efficient release of stress through fracture propagation (Figure 3b).

Similar changes in the far-field stress were observed between January 2017 and October 2017 in the vicinity of  Chasm 1. Preceding this period, the location of the rift tip remained relatively stationary for about 12 months in a transition zone between
thin (~ 100 m) marine ice and a band of regularly spaced blocks of thicker (~ 150-200 m) meteoric ice (King et al., 2018). Yet, GPS stations located on both sides of the rift indicated a slowly accelerating increase in its aperture (De Rydt et al., 2018) due to the rotation of the ice downstream of Chasm 1 towards the west and away from the remaining shelf (see Figure 1b). The period of increasing torque and slow lengthening coincided with a build-up of tensile deviatoric stress within the band of meteoric ice ahead of the rift tip (Figure 3c and d) with values estimated up to 110 kPa. In 2017, a phase of rapid propagation
followed, the onset of which was detected in January 2017 (De Rydt et al., 2018). By late October 2017, Chasm 1 had lengthened by about 4.5 km (Figure 3e), as it zipped along the boundary between an elongated, 4 km-long block of meteoric ice, and surrounding marine ice. At the same time, a noticeable reduction in the far-field stress can be seen in Figure 3e.

For both periods, we interpret the results as discontinuous (or episodic) rift propagation controlled by the heterogeneous structure of the ice shelf. The relatively stagnant phases occurred when the fracture tips encountered zones of inhomogeneous ice with different mechanical properties (King et al., 2018), causing a temporary fracture arrest and allowing the build-up of the far-field tensile stress. Once the tension caused favourable conditions for rift propagation, a phase of rapid lengthening and

stress release followed. Results suggest that discontinuous rift propagation can be expected for all Antarctic ice shelves with heterogeneous properties, and unknown spatial variations in mechanical properties of the ice can lead to significant uncertainties in the timing of fracture initiation and the speed of rift propagation.

Additional maps of tensile deviatoric stress (not shown) indicate that by October 2018, the accumulated damage to the BIS

had resulted in a significant loss of mechanical coupling between the grounded ice at the MIR and the upstream ice shelf. This loss of mechanical contact provides an explanation for the overall reduction in compressive and tensile stress across the ice shelf (Figure 3b and e). In the near future, the details of the newly emerging ice shelf configuration will depend on the exact pathways of rift propagation (De Rydt et al., 2018, Hodgson et al., 2019). In the most likely scenario, the ice shelf will approach its pre-2000 configuration with a (close-to) freely floating ice tongue, hence completing a 50-year calving cycle that started

after the last calving event in the 1970s. However, the potentially complex interaction between two active rifts at the MIR, and the nascent loss of the largest area of ice since records began in 1915, result in an uncertain future for the ice shelf.

## 6 Transient simulations of ice dynamics changes

Based on the available observational data, we identified two characteristic phases in the life cycle of the BIS: ice-shelf growth causing stress accumulation and slow-down, followed by rift formation causing stress release and speed-up. Both phases are

thought to be representative for many present-day buttressed ice shelves in Antarctica, and it is imperative that time-evolving (transient) numerical simulations of ice flow are able to represent both phases with confidence in order to make robust projections of Antarctica's future ice-shelf extent and flow.

In order to verify the capability of state-of-the-art ice flow models to reproduce observed changes in flow speed of the BIS,

we used the ice flow model Úa in a transient mode. The model simulation was started from the 01/01/1999 ice shelf configuration, with estimates of the rate factor $A(\vec{x})$ obtained from the corresponding inverse step to ensure an optimal fit between the initial model velocities and observations, as shown in Figure 4a. The initial ice draft did not make contact with the seabed at the MIR, but in order to allow the ice front to advance beyond its initial location and establish grounding at the MIR, two modifications were added to the model configuration: 1) The unknown shape of the bedrock at the MIR was

prescribed by a 3D Gaussian bump with peak elevation of 130 m below sea level, i.e. between 10 and 50 m above the local ice draft. 2) The computational domain was artificially extruded into the open ocean towards the north of the BIS, and covered with a thin layer of ice with a uniform thickness of 1 m and a spatially constant rate factor $A = 3.5 \times 10^{-25} \, \text{s}^{-1} \, \text{Pa}^{-3}$, corresponding

to ice at -10 °C (Cuffey and Paterson, 2010). The thin ice cover, which is masked in Figure 4, has limited effect on the initial dynamics of the ice shelf. A fully implicit time integration with streamline upwind Petrov-Galerkin method and stabilization (SUPG) was used, and the ice front was found to advance with limited diffusion or spurious oscillations.

After 10 years of transient evolution, during which the ice shelf geometry, ice thickness and flow velocities were allowed to freely evolve, the magnitude and spatial distribution of simulated changes in surface speed remained largely consistent with observations (Figure 4b). In particular, growth of the ice shelf generated an expanding area of ice-bed contact at the MIR and the increasing amount of basal traction, parameterized by a Weertman sliding law as described in section 4, caused a slow-down of the ice shelf by up to 1.2 m/d, both in the observational data set and the numerical simulations. The striking similarities

between the observed and modelled patterns of change between 1999 and 2010 provide a powerful validation for the predictive skill of Úa (and, consequently, for models with a comparative representation of ice dynamics) over the given time period. To our knowledge, this is the first successful hindcast of a numerical ice-flow model against observed transient changes in ice-flow velocities of an Antarctic ice shelf.

It is important to note that throughout the transient simulation, the initial spatial distribution of the rate factor was kept fixed in space, and any changes in ice flow that could result from the advection of $A$ with the ice, or changes due to temperature variations and fracture, were ignored. This approach is commonly used in transient ice flow modelling (see e.g. Arthern and Williams, 2017, Yu et al., 2018, Martin et al., 2019 for recent studies), and is based on the assumption that spatiotemporal changes in $A$ are sufficiently slow and do not significantly affect the solution on the timescales under consideration. The

agreement between observed and modelled flow changes for the BIS (Figure 4b) demonstrates that, at least between 1999 and 2010, potential changes in $A$ are not required to explain the observed slow-down of the ice shelf, and the large-scale dynamics of the BIS is, to first-order, controlled by the amount of pinning at the MIR.

However, following the re-activation of Chasm 1 in 2012 and the formation of the Halloween Crack in 2016, the assumption

of a constant rate factor $A$ breaks down. In order to capture the dynamical impact of rift formation, areas of soft ice or discontinuities in the mesh need to be introduced (see Appendices A and B for more details). Both methods provide an effective way of describing the initiation and propagation of fractures, often referred to as 'damage' (e.g. Borstad et al., 2012) or a 'calving law'.

In order to quantify the errors in numerical simulations caused by the absence of a suitable dynamical description of fractures in our model, we continued transient simulation with Úa with constant $A$ for another 8 years (2011 to 2018), and compared model output to direct observations of the surface velocity in October 2018. Between 2011 and 2018, Chasm 1 and the Halloween Crack propagated as shown in Figure 4c, and caused a loss of buttressing and widespread speed-up of the ice shelf. However, numerical projections of the flow remained largely constant or slightly decreased over this period (Figure 4c). As a

consequence, model simulations in the absence of a suitable fracture model underestimated the flow speed upstream of Chasm 1 by up to 25%, and by 100% on sections that became partly disconnected from the main ice shelf, over a period of only 7 years. The use of a constant rate factor therefore requires careful consideration and, at least for the BIS, a suitable treatment of fractures is needed to capture dynamical changes during a full cycle of growth and collapse.

## 7 Concluding remarks

Our results, based on observations and numerical modelling, demonstrate how ice shelves that are dynamically constrained by local pinning points, such as the Brunt Ice Shelf, can experience significant changes in internal stress over decadal timescales, due to their naturally evolving geometry. Favourable conditions for rifting can develop far upstream of the ice front, which makes these ice shelves particularly vulnerable to a loss of structural integrity. In combination with an often-heterogeneous internal ice structure, the mechanical conditions that control rift formation and propagation are complex and are not generally exploited in present-day ice flow models, despite recent progress (Levermann et al., 2012; Borstad et al., 2012). Existing calving criteria based on a maximum ice thickness, such as the marine ice-cliff instability mechanism (De Conto and Pollard, 2016), remain controversial (Edwards et al., 2019) and might not be directly relevant for thin floating areas such as the Brunt Ice Shelf. Other commonly-used calving laws based on minimum ice thickness criteria discard variations in mechanical properties of the ice, and are independent of internal stress. Existing theories for the vertical propagation of surface and basal crevasses (Hughes, 1983, van der Veen, 1998a, 1998b), often linked to surface hydrology (Scambos et al, 2000, Scambos et al., 2009, Nick et al., 2013), do not generally include criteria for the initiation and horizontal propagation of full-depth rifts. Glaciological changes on the Brunt Ice Shelf have unequivocally demonstrated that detailed knowledge about local pinning points, the internal structure of the ice shelf and a comprehensive treatment of fracture mechanics in ice flow models, are equally essential to capture rapid and large-scale changes in ice shelf dynamics, and thereby incorporate the critical role of ice shelves as a buffer against future mass loss from the Antarctic Ice Sheet.

## Code and data availability

All satellite data are available through the ENVEO Cryoportal (cryoportal.enveo.at); the source code of the ice flow model Úa is available from https://github.com/ghilmarg/UaSource; raw model output for the inversions (Sections 4 and 5) and the transient simulation (Section 6) are made available through the UK Polar Data Centre (DOI tbc). All other requests for data and model outputs should be addressed to J.D.R. (jan.rydt@northumbria.ac.uk).

**Author contributions**

J.D.R. and G.H.G. designed and initiated the project; T.N. and J.W. processed the satellite data; G.H.G. and J.D.R. were responsible for in-situ data collection, processing and quality control; J.D.R. performed the model simulations, carried out the data analysis and prepared the figures and manuscript; G.H.G., T.N. and J.W. reviewed and edited the manuscript.

5  **Competing interests**

The authors declare no competing interests.

**Acknowledgements**

We are grateful to the editor, Joseph McGregor, for his comments and prompt handling of the manuscript. We would sincerely like to thank both reviewers, Joe Todd and Jeremy Bassis, for their dedicated reviews with insightful and detailed comments, 10  which greatly contributed to this work.  We acknowledge the UK's Natural Environmental Research Council for providing us with the in-situ GPS data. T.N. and J.W. acknowledge support from the European Space Agency (ESA) through the ESA Antarctic Ice Sheet CCI program and from ASAP (Austrian Space Application Programme).

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

**Figure 1: Map of the Brunt Ice Shelf with inset showing its location in relation to the Antarctic continent (Howat et al., 2019). Panel a shows the ice shelf in 2010, prior to rifting. The grounding lines (solid black lines, (Bindschadler et al., 2011)), open ocean shaded in blue, and surface velocities from the MEaSUREes 2010-2011 annual Antarctic velocity map (white arrows, (Mouginot et al., 2017)), are overlain on a Landsat-7 panchromatic image collected on 4 January 2010. All velocity maps in this study have been cross-calibrated to data from a network of up to 15 in situ GPS stations. The configuration of the network in 2010 is shown by the yellow dots, with corresponding velocity arrows in black. Panel b shows the extent of two active rifts – 'Chasm 1' and the 'Halloween Crack' – in October 2018, with cyan arrows indicating their direction of propagation. Black arrows represent velocity anomalies between 2010, prior to rift initiation, and February 2018 (Sentinel-1 data), showing a dramatic increase in flow as a result of ice-shelf rifting. Blue-to-red colors illustrate the corresponding change in surface speed. Ice front locations in 1978, 2010 and 2018 are shown for reference.**

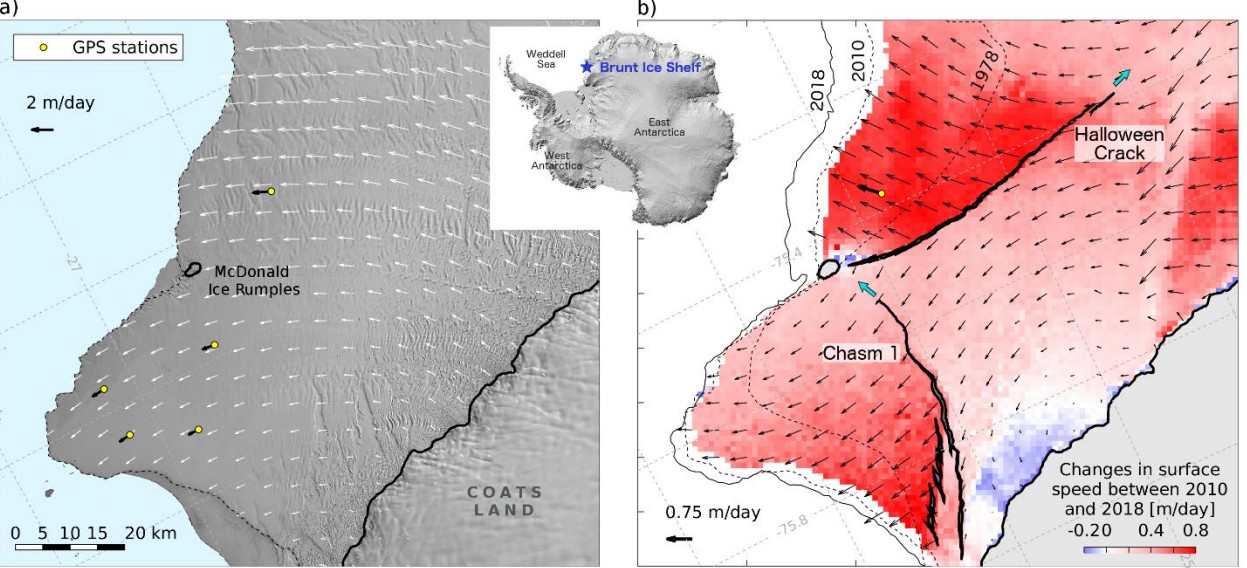

**Figure 2: Temporal evolution of principal deviatoric stress components (black arrows for extension, red arrows for compression) and maximum deviatoric stress amplitude (colours) as the Brunt Ice Shelf re-grounds at the McDonald Ice Rumples (panel b). The blue marker in panel c indicates the historical tip of Chasm 1, which corresponds to the onset location of rift propagation in December 2012. The blue marker in the panel d shows the onset location of the Halloween Crack on 4 October 2016. Panels are dated with the time stamp of the corresponding surface velocity used in the diagnostic calculation of the stress field (Table 1). Black boxes in panel d indicate the geographical extent of panels in Figure 3.**

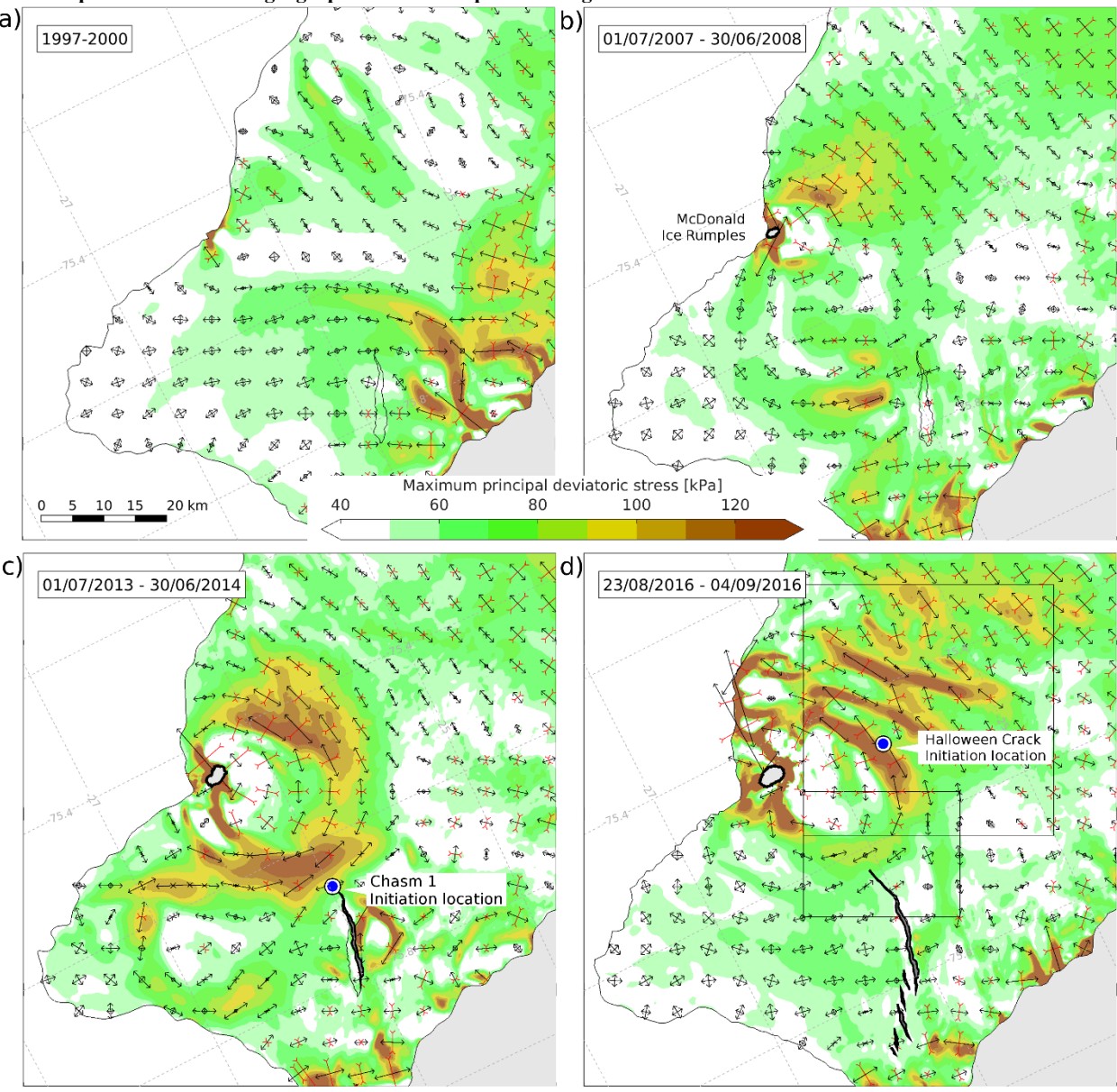

**Figure 3: Build-up and subsequent reduction in tensile stress during discontinuous rift propagation. Top row: the Halloween Crack remained stagnant for most of November 2016 resulting in the localized accumulation of stress, before propagating 11 km in December 2016 and causing a significant release of stress. Bottom row: Chasm 1 lengthened by only 500 m in 2016 compared to 1.5 km/yr in preceding years, and by January 2017, a zone of high tensile stress developed ahead of the rift tip (panel c). This zone intensified by May 2017 (panel d) and tension dissipated by October 2017 (panel e), following a rapid progression by 4.5 km. For reference, the blue markers indicate the location of rift initiation as in Figure 2, and the dashed contours panels b and e correspond to the stresses before propagation.**

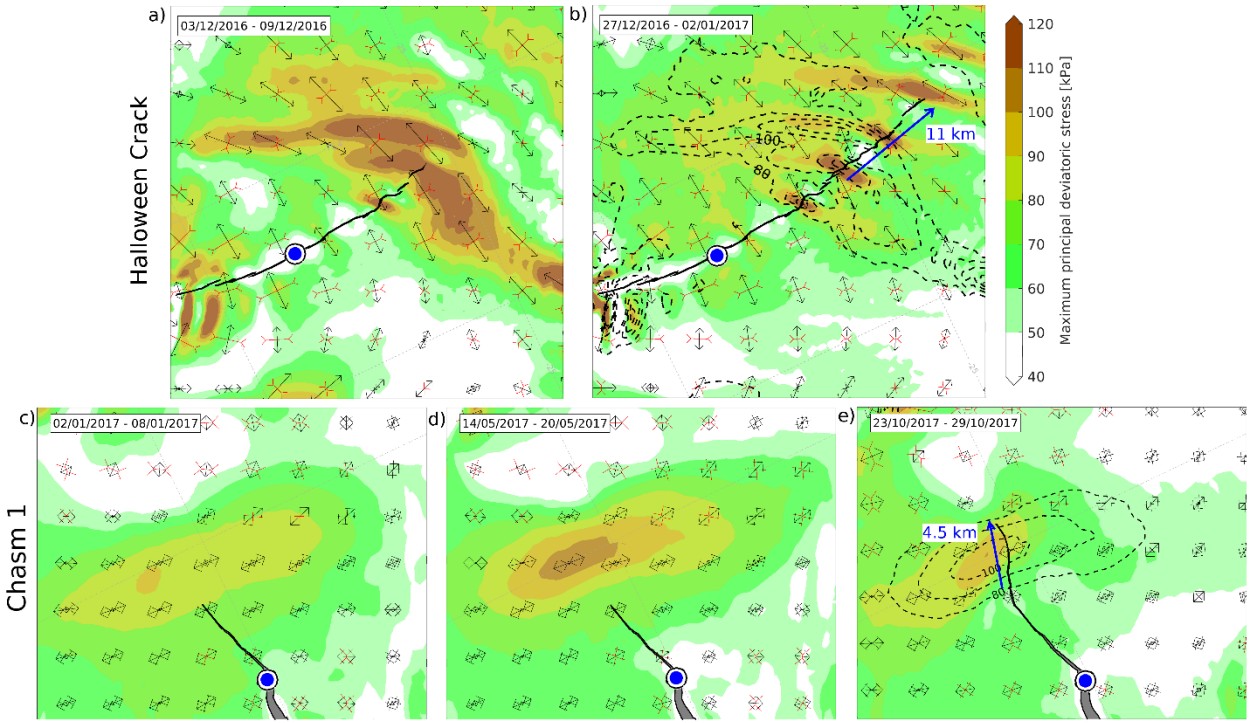

**Figure 4: Comparison between observed (left column) and modelled (right column) surface speed of the Brunt Ice Shelf between 2000 and 2018. The 2000 ice front location is shown by the dashed lines and the extent of the McDonald Ice Rumples is shaded in grey. Observations and model simulations broadly agree in 2000, and both show a significant slow-down between 2000 and 2011 due to increasing contact between the ice-shelf draft and a seabed shoal at the McDonald Ice Rumples. However, observations and model simulations strongly diverge after the formation and propagation of Chasm 1 (2012) and the Halloween Crack (2016). This difference is because Chasm1 is not generated within the numerical model due to the model's lack of a fracture mechanical component. This situation is typical for current generation of large-scale ice shelf models. Here, these differences lead to ice flow speed being underestimated by more than 1 m/day (or up to 100%) at the end of a transient run over less than a decade.**

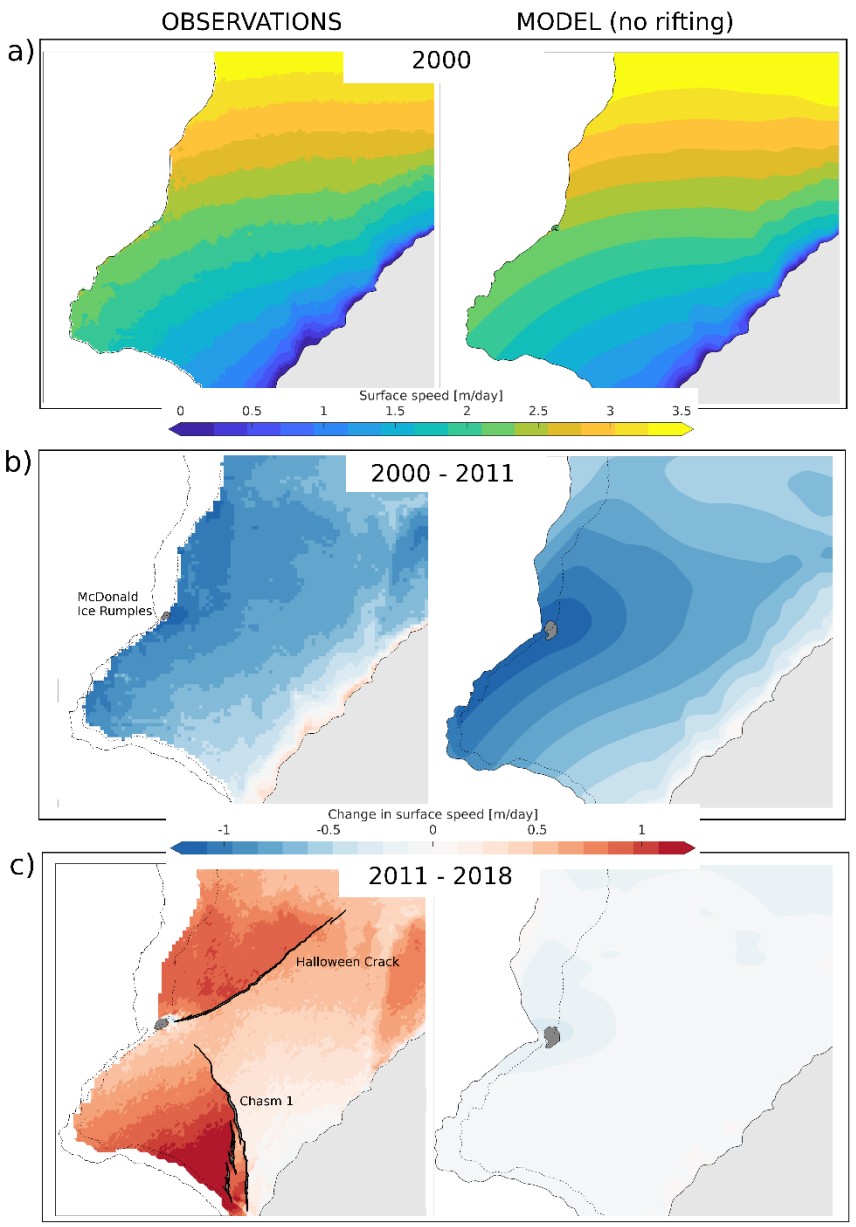

**Table 1. Data sources and corresponding timestamps used for the stress calculations. The effective timestamp in the first column corresponds to the middle of the velocity feature tracking window, and WorldView-2 surface elevations were shifted to the corresponding effective time stamp.**

| Effective time stamp | Surface DEM | Surface velocity | Ice front location |
|---|---|---|---|
| 01/01/1999 | Bedmap 2 *(Fretwell et al., 2013)* | RADARSAT1 *(Khazendar et al., 2009)*<br>1997-2001 | Landsat 7<br>14/02/2001 |
| 01/01/2008 | WorldView-2 | MEaSUREs *(Mouginot et al., 2017)*<br>01/07/2007 – 30/06/2008 | Landsat 7<br>18/12/2007 |
| 01/01/2014 | WorldView-2 | MEaSUREs<br>01/07/2013 – 30/06/2014 | Landsat 7<br>04/01/2014 |
| 29/08/2016 | WorldView-2 | Sentinel-1A/B<br>23/08/2016 – 04/09/2016 | Landsat 8<br>29/09/2016 |
| 06/12/2016 | WorldView-2 | Sentinel-1A/B<br>03/12/2016 – 09/12/2016 | Landsat 8<br>09/12/2016 |
| 30/12/2016 | WorldView-2 | Sentinel-1A/B<br>27/12/2016 – 02/01/2017 | Landsat 8<br>01/01/2017 |
| 06/01/2017 | WorldView-2 | Sentinel-1A/B<br>02/01/2017 – 08/01/2017 | Landsat 8<br>01/01/2017 |
| 17/05/2017 | WorldView-2 | Sentinel-1A/B<br>14/05/2017 – 20/05/2017 | Landsat 8<br>15/03/2017 |
| 27/10/2017 | WorldView-2 | Sentinel-1A/B<br>27/10/2017 – 29/10/2017 | Landsat 8<br>25/10/2017 |

## Appendix A. Inverse method and results

A.1 Model domain and computational mesh

The computational domain includes the Brunt Ice Shelf and Stancomb Wills Glacier Tongue, analogous to Gudmundsson et al, 2017 and De Rydt et al., 2018, in order to fully account for the weak mechanical coupling between both ice shelves. Only results for the Brunt Ice Shelf are presented here. The ice front location and extent of the McDonald Ice Rumples (MIR) for each ice-shelf configuration were outlined from satellite images, as specified in Section 3 and Table 1. The location of the southern grounding line, which marks the edge between the ice shelf and the adjacent Coats Land (Figure 1), was obtained from (Bindschadler et al., 2011). The computational domain was truncated at the grounding line, and Dirichlet boundary conditions were used to impose the velocities along this edge.

For each ice shelf geometry, an unstructured computational mesh was generated using MESH2D (Engwirda, 2014), and consisted of linear elements with 6 integration points and a mean nodal spacing of 325 m with local mesh refinement down to 100 m nodal spacing around the MIR. All results presented in the main part of the paper were obtained for a continuous mesh, and rifts were treated as 'soft ice' with a finite ice thickness. Alternatively, known rifts can be outlined from satellite imagery and cut out of the computational domain. The resulting holes in the mesh are filled with water, and have ocean pressure acting on the edges. The differences between both methods are discussed in more detail in Appendix B.

A.2 Inverse method

An adjoint method was used to obtain optimal estimates of the rate factor $A(\vec{x})$ for given surface velocities $u_{observed}$, ice thickness and ice shelf geometry. The cost function $J$ was defined as

(A1)    $J = J_{misfit} + J_{regularization}$

$$= \frac{1}{2\mathcal{A}} \iint dx \ (u_{model} - u_{observed})^2/\varepsilon^2 + \frac{1}{2\mathcal{A}} \iint dx \ \left( \gamma_s^2 (\nabla \log_{10}(A/\hat{A}))^2 + \left( \log_{10}(A/\hat{A}) \right)^2 \right),$$

with $\mathcal{A} = \iint dx$, $\varepsilon$ the data errors and $\hat{A} = 1.146 \times 10^{-8}$ kPa$^{-3}$yr$^{-1}$ the a priori value of the rate factor, which was set to a uniform ice temperature of -10°C (Cuffey and Paterson, 2010). The adjoint method calculates $A(\vec{x})$ as a solution of the minimization problem d$_A J$=0 using an iterative optimization algorithm. The algorithm was stopped after 10,000 iterations, when fractional changes to the cost function were less than $10^{-5}$. An optimal value for the Tikhonov regularization multiplier $\gamma_s$ in the cost function was determined using an L-curve approach. Figure A1 shows that $\hat{\gamma}_s$= 50,000 m produces the smallest misfit between observed and modelled surface velocities, whilst limiting the risk of overfitting, and this value for $\gamma_s$ was used throughout. The optimal value $\hat{\gamma}_s$ was found to be independent of the creep exponent $n$. Model inversions for different values of the creep exponent ($n$=2 and $n$=4) were carried out and results for the stress patterns (not shown) were found to be robust within the observational range of values for $n$ (Cuffey and Paterson, 2010). Inversions for $10 \times \hat{\gamma}_s$ and $\hat{\gamma}_s/10$ (not shown) did

not lead to any significant changes in the diagnostic stress patterns, and changes to the magnitude of the stress components were limited to less than 10%.

A.3 Examples of the rate factor $A(\vec{x})$

Figure A2 shows the estimated rate factor for two ice-shelf configurations: the left panel depicts $A(\vec{x})$ in 1999 before rift formation, whereas the right panel shows $A(\vec{x})$ in 2016 after the initiation of Chasm 1 and the Halloween Crack. Black contour lines represent the corresponding 'ice temperature' in °C, as defined by Cuffey and Patterson, 2010:

(A2) $\qquad T = \left(-\frac{R}{Q_c}\log(A/A^*) + T^{*-1}\right)^{-1} - 273.15$

with $A$ transformed to Pa$^{-3}$ s$^{-1}$ and $R = 8.314$ J mol$^{-1}$ K$^{-1}$, $Q_c = 6e4$ J mol$^{-1}$, $A^* = 3.5e{-}25$ Pa$^{-3}$ s$^{-1}$, $T^* = 263$. Values of $A$ and $T$ should be interpreted carefully, as they are vertically integrated quantities that do not only vary with ice temperature, but also include other effects such as ice rifting. This is obvious from the right hand panel, where consistently high values of $A$ are found along the rift trajectories and other crevassed areas such as the hinge zone immediately downstream of the grounding line. These areas of 'soft ice' accommodate the high strain rates or discontinuities in flow speed in those areas (compare to Figure 1b). At the MIR, extreme values of $A$ can also result from fitting the data to the SSA flow approximation, which breaks down here because of the high vertical shear. In both panels of Figure A2, bands of stiffer (colder) ice are seen to follow flowlines from the grounding line to the ice front, and have previously been identified as bands of meteoric ice that originate upstream of the grounding line, in contrast to the surrounding areas that predominantly consist of (warmer) marine ice (King et al., 2018). The recovery of the internal ice structure from $A$ provides both an independent confirmation for the work of King et al., 2018, which was based on ground penetrating radar data, and additional support for the physical meaningfulness of $A$.

**Appendix B. Representation of rifts in the computational domain.**

Rifts that cut through the full thickness of the ice shelf can be (partially) filled with ice mélange, marine ice and snow. In some cases, the infill creates a mechanical coupling between vertical rift faces and provides tensile strength, as pointed out by Larour et al., 2004 for rifts in the Ronne Ice Shelf. The use of a continuous computational mesh in the inversion, which allows for non-zero ice thickness inside the rifts, seems most appropriate in this case. On the other hand, open water leads have routinely been observed inside rapidly-evolving rifts such as Chasm 1 and the Halloween Crack, and opposite vertical rift faces are not or only partially connected. This justifies the representation of rifts as holes in the mesh, with ocean boundary conditions applied to the edges (the Hybrid and Water experiments in [Larour et al., 2004]). In case of the BIS, a numerical perturbation experiment by Gudmundsson et al., 2017 has shown that, prior to the reactivation of Chasm 1 in 2012, its mélange-filled area could be removed from the computational domain and replaced by open water without significant instantaneous impact on the dynamics of the ice shelf.

In general, mélange thickness and areas of open water are not well constrained by observations, and the most appropriate choice of mesh type (continuous or with holes) is unclear. Here we demonstrate that, at least for Chasm 1 and the Halloween Crack, the rate factor and diagnostic stress distribution are not critically dependent on this choice. In Figure B1 we compare values of the inferred rate factor $A$, the misfit between observations and model velocity, and the diagnostic principal stress components for the 06/12/2016 ice-shelf configuration. The left panels show results for a continuous mesh with a mélange thickness extrapolated from the thickness of neighbouring ice shelf areas. On the right, elements corresponding to rifts in the ice shelf were removed from the mesh, and ocean boundary conditions were imposed along the newly exposed faces.

For both limiting cases, the misfit between the modelled and observed flow speed is largely comparable (see insets in Figure B1) and relative errors are on the order of 5% or less. For a continuous mesh, high values of the rate factor along the rift trajectories represent weak ice, and reflect discontinuities in flow speed (or high strain rates) across the rifts, whereas such high values are mostly absent when rifts are represented by open water. In the latter case, any remnant areas of weak ice along the rifts are likely due to discrepancies between the outlines traced from visible satellite images and the true extent of the active rift (De Rydt et al., 2018). The principal stress directions are very similar in both cases, but with some notable differences in the magnitude of the maximum principal stress, in particular close to the tip of the Halloween Crack. The misfit between observed and modelled velocities in this area is somewhat larger in the open water case compared to the mélange case, causing a less accurate fit of the model to the observed strain rates, and a lower confidence in the derived stresses. All results in the main part of this paper were based on a continuous mesh.

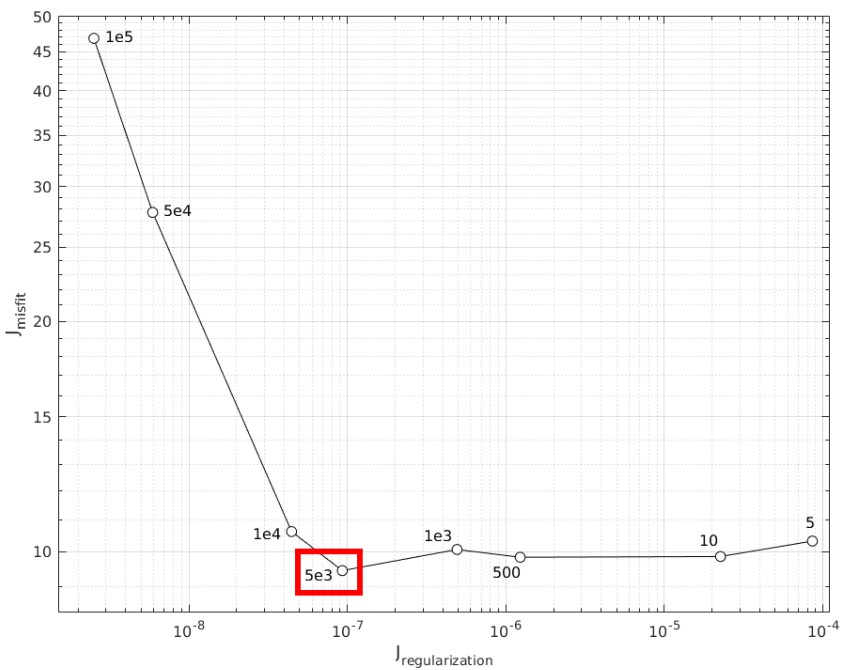

**Figure A1. Example L-curve for the 01/01/2014 ice shelf configuration.**

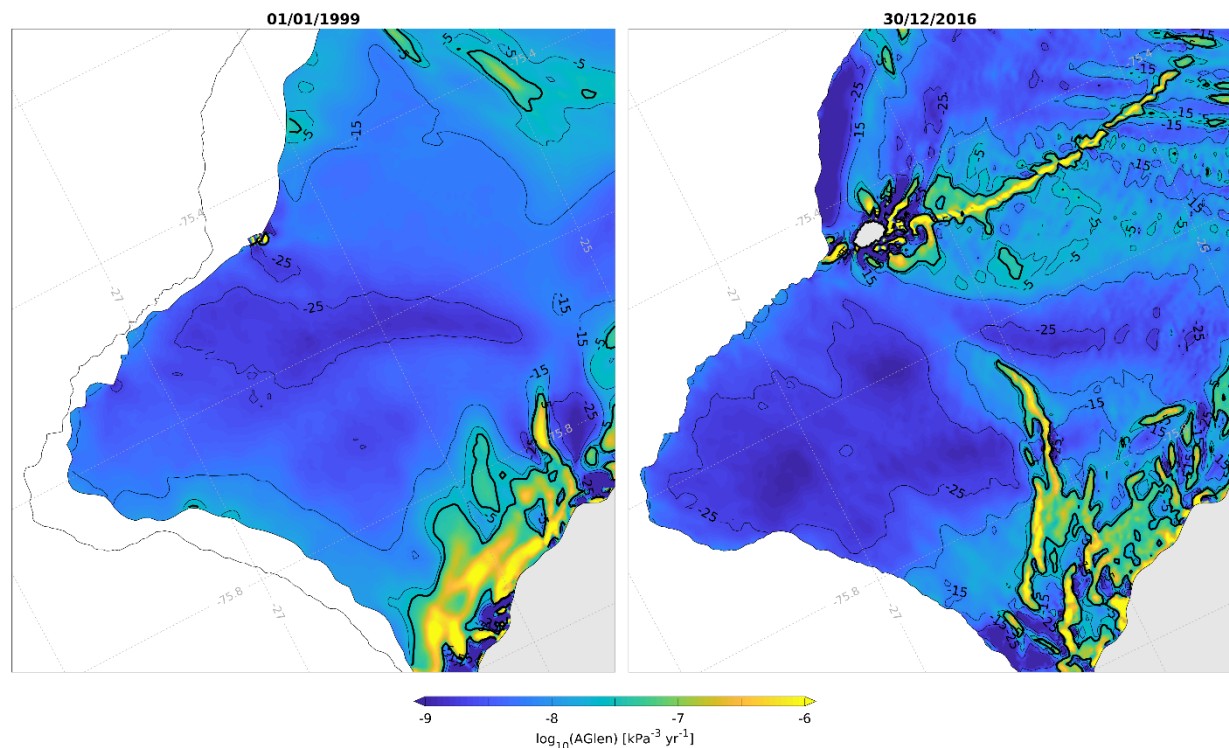

**Figure A2.** Examples of the rate factor *A* (colours) and associated 'ice temperatures' in °C (black contours) as calculated from eq. (A1) for the Brunt Ice Shelf prior to rift formation (01/01/1999, left panel) and after the initiation of Chasm 1 and the Halloween Crack (30/12/2016, right panel). Contours are plotted at 10 °C intervals and the zero degree contour is highlighted by the thicker line. The left panel shows the 2016 ice shelf extent for reference.

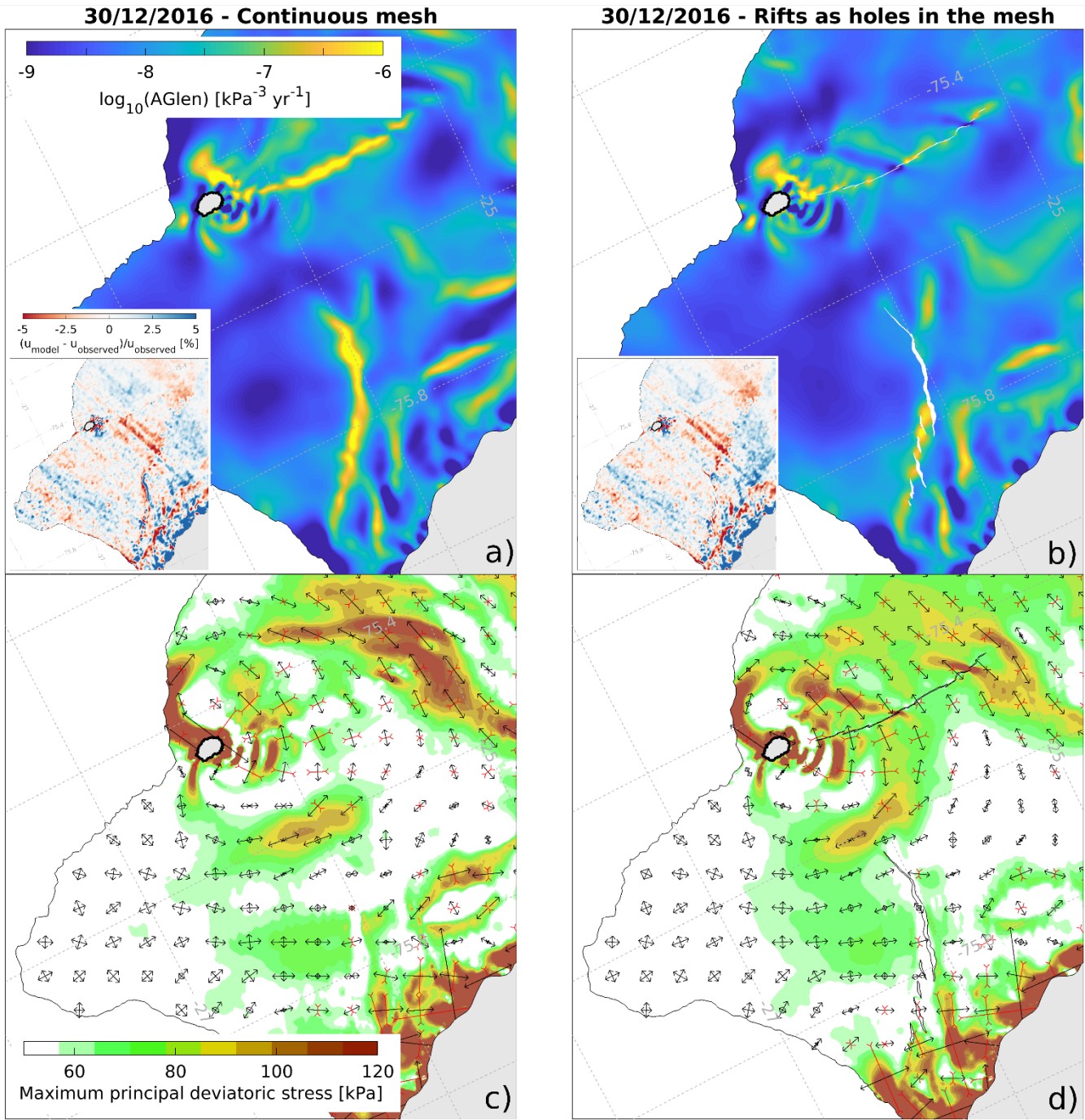

**Figure B1. Spatial maps of the rate factor *A* (panels a and b) and diagnostic principal stresses (panels c and d) for the Brunt Ice Shelf on 06/12/2016. Results are based on identical input datasets, but for two different computational meshes: on the left (panels a and c), a continuous mesh was used and Chasm 1 and the Halloween Crack were filled with ice; on the right (panels b and d), rifts were represented as holes in the mesh with ocean boundary conditions.**