# Peer review of "Calving cycle of the Brunt Ice Shelf, Antarctica, driven by changes in ice-shelf geometry"

_The Cryosphere, 2019_

## Referee Comment (RC1) · Joe Todd (Referee) · 29 Apr 2019

**Review of "Calving cycle of the Brunt Ice Shelf, Antarctica, driven by changes in ice-shelf geometry" by De Ryt et al. - Joe Todd**

This study combines multiple observational records with inverse modelling to study the ice dynamics, stress & fracture of the Brunt Ice Shelf. Data from satellites & in situ measurements are assimilated into the SSA model Úa to invert for the flow parameter A across the shelf, and the resulting stress maps are analysed to build up a timeline of ice shelf stress conditions before, during and after the re-activation of Chasm 1 and the propagation of the Halloween crack. This is an interesting and well-presented study which warrants publication in The Cryosphere; as the authors note, the 'natural' cycle of stress concentration and release on ice shelves is a major factor controlling calving. I strongly agree with the conclusion that full-thickness rifting should be resolved in ice-sheet models. I think the manuscript could benefit from some additional details on the modelling results and some clarifications.

**General comments:**

It is not totally clear from the figure captions & text whether the stress maps shown in Figs. 2 & 3 come from Úa model output. I can see 3 possibilities: (1) The stress maps are produced using observed velocity (from which strain can be derived) and an assumed constant flow parameter A. (2) As above, but A comes from Úa output. (3) The stress maps are a direct output of Úa simulations. The text strongly implies (3) but, from reading the methods section, I do not think that any fractured domains were studied with the model. Is rifting accounted for through inversions (i.e. low A where rifts exist)? If (3) is the case, more details should be added to explain how the rifting is accounted for. If (3) is not the case, clarifications and modifications are required in the text to avoid giving a false impression to the reader. It would be nice to see the results of the model inversion (maps of 'A') and this would probably also help clarify the point above. In general, it's just not very clear at present exactly *how* the model was used.

**Specific Comments:**

P3L9-10 – Can the broad-scale pattern of ice shelf thinning be established? Paolo et al. (2015) seem to provide data which covers the BIS. You make a compelling argument for the first-order importance of internal dynamics/heterogeneity for crack propagation, but does this completely preclude any external environmental signal?

P4L3: This sentence implies that all the stress results shown in the paper are Ua model output. Is that the case?

P4L8: Could you show the inverted-for A parameter, perhaps in supplementary material? Presumably there are some pretty interesting patterns.

P5L3-11: I am not totally clear what the approach is here. How do you shift the DEM to an effective timestamp? What does the LIDAR data provide?

P6L14: Again, this strongly implies that Fig 2 & 3 represent model output.

P6L26: 'Ocean pressure acting on newly formed rift surfaces' - I'm slightly confused by this. On a floating shelf, the overall ocean pressure should be equal to the cryostatic pressure which existed before the crack formed. The exception, which I guess applies here, is if the intact shelf was under significant tension. But is it really accurate that the ocean pressure is pushing the rift apart? I'd have thought that its the concentration of the supported stresses onto a narrower band (the remaining intact ice) which promotes further fracture growth.

Fig 1 or 2: As I was reading the results section, I was thinking it'd be nice to clearly visualise the compressive arch. Could you perhaps add a panel (or overlay Fig 1a) showing regions with extension in both directions, versus one compressional component (like Doake et al., 1998).

Fig 4b: Observations & model match well to the south of the MIR, but the difference grows further north. Can you speculate why?

**Minor Comments:**

P2L27: A bit pedantic, but I think 'single' would be better than 'singular' here. 'Singular' tends to refer to an exceptional event or thing.

P3L15: 'preconditions for rifting were re-established'. What were these preconditions? I think the rest of the paper lays out what these preconditions were, but its perhaps a little premature to say this here without explanation.

P3L18: 'singular' again

P4L11: slight formatting error – ref in brackets

Fig 1: North arrow?

Fig 2: Unless is really reduces the clarity of the figures, I'd think that for a colour scale with a white minimum, the minimum ought to be 0 kPa.

---

## Referee Comment (RC2) · Jeremy Bassis (Referee) · 22 May 2019

**1   Overarching comments**

This study describes a comparison between observations and model inferred stresses for the Brunt ice shelf. The authors demonstrate that the collision of the Brunt ice shelf with a pinning point resulted in increased compressive and tensile stresses and argue that this resulted in the increased tensile stresses needed to reactivate rift propagation. Overall, the results are highly glaciologically relevant and add to our understanding of ice shelf rift propagation.

The manuscript does feel a little bit like it was originally intended for a short form journal

with a restrictive length requirement and has not taken advantage of the more generous space allocated by longer form journals. As such, I had a hard time understanding what was actually done and it seemed as though there were critical details missing from the exposition. Without those details it is hard to assess the reliability of the methods and conclusions drawn. There are also a few places in the manuscript where the text does not appear to accurately reflect the literature or conclusions are not fully supported by the results. My review is relatively long, but most of the issues are (hopefully) easy to correct by expanding the text to include more critical information.

**2 Major Comments:**

The most significant issue in the manuscript relates to how the authors define a "rift" in the model and how this compares to how the rift actually behaves. We know that rifts in ice shelves can be discontinuities in the ice (fractures) that are filled with ocean water. Rifts are also often filled with a mixture of snow, sea ice and blocks of ice called melange. The melange can become structurally coherent to the point that rifts are barely visible on the surface of the ice, although this tends to be more common in relic rifts that have not been active for decades (or even centuries). Rifts can be represented in models in different ways. The most physically consistent way is to incorporate rifts as actual discontinuities in the ice shelf (see Larour et al., 2014). One then needs to account for the thickness of melange that fills rifts when applying a normal traction boundary condition along the rift walls, analogous to the calving front boundary condition. Historically, rifts have also been represented in models as regions of intact ice with a rate factor that is set (or inferred) to be much lower than is traditional for intact ice. In this representation, rifts not discontinuities and these features essentially behave like diffuse zones of really warm ice and NOT as fractures. The authors need to be clear about how rifts are represented in their model before any inferences can be made about the effect of "rifts" on the dynamics of ice shelves.

A second, and closely related point, arises from circularity in tuning a model to fit observations and then using the fact that the model fits the observations to argue that the model is appropriate. As noted earlier, there are different ways of representing rifts in a model and it is far from clear that the stress field associated with these different representations of rifts are equivalent. In fact, representing rifts as discontinuities will generate stress concentrations near the tip of the rift that will not be present in models that represent rifts as diffuse zones of soft ice. Moreover, when tuning a model to match observations, it is often possible to absorb all uncertainties and errors into the parameters that are being tuned. For example, in traditional damage mechanics, the damage affects the Cauchy stress and thus would also affect the driving stress on the right hand side of the SSA equations. Similarly, errors in density or briny layers of marine ice could also affect the driving stress. One can, of course absorb this error into inversions for the rate factor, but it is less clear that the inference of the rate factor is *physically* significant. Here, the authors can do more to make their case by describing how rifts are represented in the model and showing actual maps of the rate factor and, if possible, converting those maps to physical variables, like equivalent ice temperature. Fortunately, the Brunt ice shelf itself appears to be well studied and the authors should be able to compare inversion results with the position of known bands of marine ice inferred by King et al.

We get into similar issues when the authors argue that rift widening causes stress concentrations ahead of the rift. I don't follow the inference if all the authors have done is tune the model to match observations. The authors can, however, make this inference, if they have instead performed a forward model run and the widening of the rift predicted by the model matches observed widening rates and the stress increase near the tip of the rift matches the inferred stress increase. Again, this points to a need for some explanatory text.

Larour, E., et. al., (2014),Representation of sharp rifts and faults mechanics in modeling ice shelf flow dynamics: Application to Brunt/Stancomb-Wills Ice Shelf, Antarctica,

J. Geophys. Res. Earth Surf., 119, 1918-1935, doi:10.1002/2014JF003157.

**3 Detailed comments**

1. The introduction states that the focus of the manuscript is "on the more commonly neglected internal drivers that underlie rift initiation and propagation." It is far from clear to me that this accurately reflects the literature. For example, Fricker et al., 2002, Joughin and MacAyeal, 2005, Larour et al., 2004; Borstad et al., 2012 and 2017 all examine the glaciological stress as the dominant factor driving rift propagation. (There are many, many more citations. These are just a few examples. In fact, almost all of the literature that I can find points towards glaciological stress as the driver.)

More recently (and in the Cryosphere) Arndt (2018) note the role of pinning points in generating rifts in Pine Island glacier, which seems analogous to the study here. In fact, as far as I can tell, with the exception of the Antarctic Peninsula Ice Shelves (that all seem to have effective social media accounts and PR departments), most of the literature for ice shelves does focus on internal stresses. That said, the controls on rifting remain poorly understood, in part because of the long time scale between calving events make the process difficult to directly observe. This observational deficiency, in my opinion, is one of the more significant motivations and strengths of the present manuscript and it would be useful to re-emphasize this to readers.

Fricker, HA, Young NW, Allison I, Coleman R. 2002. Iceberg calving from the Amery Ice Shelf, East Antarctica. Annals of Glaciology, Vol 34, 2002. 34

Joughin, I., and MacAyeal, D.R. (2005),ÂăCalving of large tabular icebergs from ice shelf rift systems,ÂăGeophys. Res. Lett., 32, L02501, doi:10.1029/2004GL020978.

Larour, E., Rignot, E., and Aubry, D. (2004), Modelling of rift propagation on Ronne Ice Shelf, Antarctica, and sensitivity to climate change, Geophys. Res. Lett., 31, L16404,

doi:10.1029/2004GL020077.

Borstad, C., McGrath, D., and Pope, A. (2017), Fracture propagation and stability of ice shelves governed by ice shelf heterogeneity, Geophys. Res. Lett., 44, 4186–4194, doi:10.1002/2017GL072648.

Borstad, C.P. et al. A damage mechanics assessment of the Larsen B ice shelf prior to collapse: Toward a physically-based calving law. Geophy. Res. Lett. 39, L18502 (2012).

Arndt, J. E., Larter, R. D., Friedl, P. , Gohl, K., Hoppner, K.  ãand  ãthe Science Team of Expedition PS104, (2018): Bathymetric controls on calving processes at Pine Island Glacier , The Cryosphere, 12(6), pp. 2039–2050. doi:10.5194/tc-12-2039-2018

2. Iceberg calving is a natural process and this needs to be more clearly emphasized

The broader context in the introduction is the looming calving event from the Brunt Ice Shelf. Building on the previous point, it is known that ice shelves exhibit a natural cycle of decades to centuries advance punctuated by episodic retreat associated with the detachment of large tabular icebergs (see, Fricker et al., 2002, Walker et al., 2015). These calving events are believed to be driven by a combination of the accumulation of fractures coupled with changes in the glaciological stress. Yes, there is evidence for climate driven disintegration of ice shelves, primarily on the Antarctic Peninsula, but this is more of an exception to the norm. I suggest that the authors consider adding more context to the introduction, explaining not only that calving is part of the natural cycle of ice shelves, but how the calving event from the Brunt Ice Shelf fits into this larger context. How similar is this event from previous events? Or are are there no records of previous events? How does the cycle compare to other ice shelves?

Similarly (and sorry for being pedantic), one of the issues that hindered my understanding of the broader context of the study is that the term "unique" is used frequently (4 times at least) and it was unclear what, exactly was unique in each of these instances?

The first time unique was used, it was used to describe the 50-year time series. This seems like appropriate usage. But, the next time we are told there is "a unique opportunity to enhance . . . process-based understanding". What is unique about the opportunity? Is this the 50-year time series? If this calving event is a continuation of the natural cycle, then (pedantically), the opportunity is not unique. There are also other rifts on other ice shelves that have been (or can be) studied. What exactly is unique about this opportunity/rift? The third time we are told there is a "unique, network of up to 15 GPS". GPS have been deployed around rifts (propagating and not propagating) in ice shelves multiple times so what about this particular deployment is unique. Finally, we are told "BIS represents a unique setting . . . calving processes can be studied" This sounds like the authors are arguing that rift propagation/iceberg calving is different in this situation than the calving cycle that is observed elsewhere? It would be helpful to clarify all of this.

Overall, I think that the authors could help readers understand the significant of the Brunt Ice Shelf and the particular rift system by sketching out what is common about the iceberg calving process across ice shelves. Then, tell us what is unusual in this situation (is it just the observations? the pinning point?) and what is truly unique here (is Brunt itself unique due to the large heterogeneities?). This would enable readers to better understand how this study fits into the broader context of rifting and calving from other ice shelves.

Walker, C., Bassis, J., Fricker, H., Czerwinski, R. (2015). Observations of interannual and spatial variability in rift propagation in the Amery Ice Shelf, Antarctica, 2002–14. Journal of Glaciology, 61(226), 243-252. doi:10.3189/2015JoG14J151.

3. Methods, part 1 (inversions)

This is the where I really started to struggle to understand what was done and there is critical detail missing from the description of the model and inversion process. The model is described as a shallow-shelf approximation model SSA, which is standard

for ice shelves. My understanding of the inversion is that the authors invert for the rate factor A(x,y) by ingesting surface velocities into the model. However, the inversion uses two regularization parameters $\gamma_a$ and $\gamma_s$ neither of which are defined in the text. Digging into Reese et al., 2018, it looks like the regularization parameters correspond to the ice softness AND basal friction coefficient. But an ice shelf, by definition, is freely floating and there is no basal friction. Are the authors inverting for basal friction beneath the pinning points? Is this done everywhere or in certain places? More details and more clarity are needed to understand what has been done here. I now see all the way at Page 8 that a Weertman sliding law is used specifically for the pinning points. This needs to be explained much earlier if it fits into the inversions. Also, what is the shape of the pinning point? Is the ice shelf plowing over it or is the pinning point just tickling the bottom?

4. Deviatoric stress is not constant with depth

There are also more subtle issues associated with the interpretation of the inferred rate factor. In the SSA approximation strain rate is independent of depth. However, the stress is only independent of depth if the temperature in the ice is constant within each column of ice. What the authors are really inferring is the depth averaged rate factor. A consequence is that the authors are also only able to show the depth averaged deviatoric stress. Stress could be much higher near the surface of the ice, where temperatures are likely much colder. This needs to be recognized and explained and in particular, related back to the physical interpretation of the rate factor of ice.

5. Methods, part 2 (maps of rate factor please)

It helpful to readers to see the actual maps of inferred ice softness and basal friction (if this was also inverted for). This would certainly help convince readers that patterns of rate factor are realistic and not spurious artifacts. This is a matter of preference, but I personally also like to see the inferred rate factor converted into an ice temperature so that we can be sure that the ice temperature is semi-realistic based on known conditions. The authors note that these are related to structural properties of the ice. In particular, it would be helpful to know if the inversions for ice softness correspond to regions of marine ice documented by King et al., (2018). In fact, one also wonders if the inversion could resolve the sharp variations in material properties associated with the bands of marine ice documented by King et al., (2018). A standard way to test this is by doing a "checkerboard" test. You compute the forward model using a checkerboard or other pattern. You then add noise to the signal and invert based on the synthetic data. This would give a sense of the resolution of the inversion and if the inversion can pick up relevant structural features. The more formal way of doing this would be to construct resolution kernels to formally determine what can and cannot be resolved.

6. Observations, how do you shift the data to a date?

I thought the observation section was much clearer and easier to understand. But it was unclear to me how you "shift" a DEM to an effective time step? There is no reference or description of the method used to do this. This, along with any error associated with the procedure should be described.

7. Can a viscous ice shelf model really accumulate stress at the tip of a rift by dissipating gravitational potential energy?

The authors argue that rift widening results in accumulation of stress ahead of the rift. The energy balance in an ice shelf model tells us that gravitational potential energy is dissipated through viscous flow. The accumulation of stress seems to imply energy is being added to the system faster than it can be removed. What is the source of the energy that is added to the system that drives energy accumulation? Is this related to torques associated with rotation of the blocks of ice? Is this conclusion supported by forward model runs or this based on tuning the model to match observations? Given the fact that rift widening is documented by GPS, it seems as though the authors should be able to do a forward model run to compare simulations with observed rift widening and use the forward model to show that stress is concentrated ahead of the rift.

**8. Conclusions**

This is up to the author, but there has been little doubt that "calving laws" are needed in ice sheet models and this is not the main conclusion I would draw from this study. I don't think it will come as any surprise to most readers that iceberg calving is an important process in ice sheet evolution. The authors also have a typo in their description of the so-called marine ice cliff instability. The marine ice cliff instability assumes that there is a **maximum** ice thickness, not a **minimum** ice thickness. Moreover, the marine ice cliff instability generally applies to thick grounded ice and not thin ice shelves, like the Brunt Ice Shelf. Minimum ice thickness models have been a mainstay in ice sheet models for decades largely as a means of preventing ice shelves from indefinitely advancing. Hence, the authors have a good argument that these minimum thickness criteria are not physical.

Actually, coming back to the Deconto and Pollard marine ice cliff instability study, my understanding is that the parameters used by Deconto and Pollard were derived based on parameter sensitivity studies for past sea level rise. These are, technically, observations are they not? Direct observations of ice flow of ice sheets hundreds of thousands of years ago, similar to the GPS and satellite imagery used in this study, remains a challenging problem in paleo ice sheet studies. And if the marine ice cliff instability is really a thing, the only evidence we have likely comes from past ice sheet conditions when these processes may have been active. Here, it would be useful if the authors put their results in context of past and future projections of ice sheet change. If structural heterogeneity is important, is it possible to predict it instead of tuning a model to match observations? How important is structural heterogeneity versus the geometry of pinning points? It seems like knowing the location of pinning points (which is possible) could at least provide a first order approach to rift generation even if it does not match the detailed sub-decadal trends? This study potentially offers a lot of information and it would be useful to readers to see how this fits into the bigger picture.

Minor comments:

Page 5, line 25: What is "Geometric Deformation"? Do the authors mean that the geometry of the ice shelf is changing? I actually googled this term, but all of the hits directed me to papers on differential geometry, which seems like it is not what the authors are talking about.

Page 6, line 26. Ice is in hydrostatic equilibrium. A force balance at the ice-ocean interface (analogous to that at the calving front) within a melange free rift suggests a deviatoric stress pointing into the rift. Why is the ocean pressure pulling the rift apart? The large scale stress of the ice shelf might pull the ice apart. This should be clarified.

Page 3, line 28: We were told there is 50 years of data, why only focus on the period from 1997-2018? What is the benefit of the long time series if less than half are used? The earlier emphasis on 50 years of data seems like a bit misleading at this point.

Page 1, line 20: This is pedantic, but I would consider the 5000 km$^2$ berg that detached from the Larsen Ice Shelf to be a small to mid-sized berg. Iceberg B15 that detached from the Ross Ice Shelf was twice as large and Shackleton documented icebergs that were even larger.

Page 1, line 20: The word "since" refers to time. For example, "It has been a long time since the Knicks won the championship." In this case, I believe you want "Because".

Page 2, line 9: The references given here document thinning of ice shelves and do not appear to describe any links between thinning and calving.

Everywhere: space between numbers and units 3m should be 3 m

Page 7, line 2: comma after "but"

Page 7, 2nd paragraph: Now I'm really confused about what is going on. Are the authors introducing a rift into the model and widening it, based on observations to examine the stress field. Or, have the authors inverted for stress (OK, actually ice softness) based on surface velocities at several intervals of time? In the first case, I think the authors are safe saying that the increase in stress is due to rift widening. In

the second case, I don't know that you can say that the stress is caused by widening when no rift widening has been included in the model and the model has been tuned to reproduce surface velocities (and hence stresses). Morever, the assumption that rift widening results in stress concentration should be checked against other periods of time when rift propagation did not occur. For example, there is a long history of rift widening without propagating prior to Chasm's reactivation. Does this period of time correspond to the rift propagating into a zone of marine ice?

Page 7, line 26: The technical jargon "damage" is introduced here. Authors should avoid the term or define it. Keep in mind that "damage" has a precise definition in the fracture mechanics literature and is, most generally, a tensor. The term damage is often used heuristically in glaciology in confusing and imprecise ways. If the authors mean rifting, I recommend just saying rifting.

Section 6, Page 8, section paragraph: Wait a minute. Why is the rate factor A(x,y) not a property of ice that advects with the ice? Conventionally, the rate factor has been linked to temperature, grain size and crystal structure of the ice. If reductions in the factor A(x,y) are linked to fractures then surely these must also advect with the ice? If the advection of the rate factor is not important, then why is heterogeneity of the ice important? I'm missing something critical here because this seems like this contradicts the authors main conclusion that heterogeneity is important.

The "extrusion" method for calving front advance is known to generate significant artifacts if not treated carefully. The calving front should advect as a sharp shock and accurate shock capturing methods are needed to avoid overly diffusing the calving front. Numerical details of advection should be included with limitations described. Does the advection scheme preserve mass? It is diffusive? Does it preserve the shock-like nature of the calving front? Are results sensitive to grid size or time step size?

Page 8, last paragraph: The statement that ice sheet models keep the calving front pinned to present day conditions might have been true a decade or two ago, but pretty

much all of the major ice sheet models at this point allow the calving front to evolve. PISM uses a wetting drying algorithm combined with "eigen calving". ISSM uses a level set method combined with a Von Mises calving law. BISICLES and CISM have their own methods to advance the calving front and use a spectrum of calving laws. These days, models allow the calving front to advance and retreat according to heuristic (and often known to be incorrect) parameterizations. Whether advancing and retreating the calving front based on inaccurate and unphysical calving laws is progress is a question that I will leave to others.

Page 12, line 12: Reference to Lipovsky, 2018b appears to reference an unpublished manuscript. Check Cryosphere style guidelines for rules on references to non-peer reviewed literature. This is prohibited by AGU publications, but the standards of TCD might not be as stringent

Figure 2-3. Best not to use a red-green color scale.

---

## Author Comment (AC1) · 28 Aug 2019

**Review of "Calving cycle of the Brunt Ice Shelf, Antarctica, driven by changes in ice-shelf geometry" by De Ryt et al. - Joe Todd**

This study combines multiple observational records with inverse modelling to study the ice dynamics, stress & fracture of the Brunt Ice Shelf. Data from satellites & in situ measurements are assimilated into the SSA model Úa to invert for the flow parameter A across the shelf, and the resulting stress maps are analysed to build up a timeline of ice shelf stress conditions before, during and after the re-activation of Chasm 1 and the propagation of the Halloween crack. This is an interesting and well-presented study which warrants publication in The Cryosphere; as the authors note, the 'natural' cycle of stress concentration and release on ice shelves is a major factor controlling calving. I strongly agree with the conclusion that full-thickness rifting should be resolved in ice-sheet models. I think the manuscript could benefit from some additional details on the modelling results and some clarifications.

Thank you for these positive comments.

**General comments:**
It is not totally clear from the figure captions & text whether the stress maps shown in Figs. 2 & 3 come from Úa model output. I can see 3 possibilities: (1) The stress maps are produced using observed velocity (from which strain can be derived) and an assumed constant flow parameter A. (2) As above, but A comes from Úa output. (3) The stress maps are a direct output of Úa simulations. The text strongly implies (3) but, from reading the methods section, I do not think that any fractured domains were studied with the model. Is rifting accounted for through inversions (i.e. low A where rifts exist)? If (3) is the case, more details should be added to explain how the rifting is accounted for. If (3) is not the case, clarifications and modifications are required in the text to avoid giving a false impression to the reader. It would be nice to see the results of the model inversion (maps of 'A') and this would probably also help clarify the point above. In general, it's just not very clear at present exactly *how* the model was used.

Snapshots of observed surface velocities and ice shelf geometries were assimilated into Úa, and for each snapshot, an optimal solution for *A* was obtained through an inverse method, which optimizes the misfit between observed and modelled surface velocities. The resulting solutions for *A* together with the diagnostic surface velocities were used to calculate the stress maps. We have clarified our methodology in section 3, and added further details about the inverse method and resulting maps of *A* in Appendix A. To address your concern about the absence/inclusion of fractures in the domain, we also provided a comparison for *A* and the stress field between (1) the inversion with a continuous mesh (i.e., rifts are filled with ice/melange), and (2) the inversion with rifts as holes in the mesh (i.e., rifts correspond to open ocean) in Appendix B. In summary, method (1) produces high values of *A* along the trajectory of active rifts, whereas in method (2) these high values are absent. Yet, the derived large-scale stress distribution of the ice shelf remains qualitatively similar between both methods, which demonstrates the robustness of our results with respect to the representation of rifts in the model domain.

**Specific Comments:**

P3L9-10 – Can the broad-scale pattern of ice shelf thinning be established? Paolo et al. (2015) seem to provide data which covers the BIS. You make a compelling argument for the first-order importance of internal dynamics/heterogeneity for crack propagation, but does this completely preclude any external environmental signal?

We have no additional data on broad-scale changes in ice thickness other than the maps produced by Paolo et al.. This work was cited in the paper.

P4L3: This sentence implies that all the stress results shown in the paper are Ua model output. Is that the case?

All stress maps were calculated from diagnostic model output, i.e., an optimal solution for *A* and corresponding surface velocities for different snapshots in time. 'Optimal' is defined in the sense of inverse theory, i.e. as a minimum of the cost function which penalizes the difference between (gradients of) the observed and modelled surface velocities (see Appendix A for further details).

P4L8: Could you show the inverted-for A parameter, perhaps in supplementary material? Presumably there are some pretty interesting patterns.

More details about the model inversion and patterns for *A* are included in Appendix A.

P5L3-11: I am not totally clear what the approach is here. How do you shift the DEM to an effective timestamp? What does the LIDAR data provide?

Both points have been further clarified in the text.

P6L14: Again, this strongly implies that Fig 2 & 3 represent model output.

The calculation of the stress field requires the surface velocities in combination with a rheological model. As a first approximation, the rheology *A* can be assumed to be spatially constant and observed velocities can be used to calculate the stress field. However, a more accurate approach is to allow for a spatially variable *A* (obtained through a formal inversion of the observed surface velocities) and use the corresponding modelled surface velocities. In the latter approach, which was followed here, the forward model can be interpreted as a physically-based filter to reduce the measurement noise. These points have been clarified in section 3.

P6L26: 'Ocean pressure acting on newly formed rift surfaces' - I'm slightly confused by this. On a floating shelf, the overall ocean pressure should be equal to the hryostatic pressure which existed before the crack formed. The exception, which I guess applies here, is if the intact shelf was under significant tension. But is it really accurate that the ocean pressure is pushing the rift apart? I'd have thought that its the concentration of the supported stresses onto a narrower band (the remaining intact ice) which promotes further fracture growth.

We did not mean to imply that ocean pressure drives rift propagation, rather to emphasize the importance of normal pressure acting on newly formed surfaces for the force balance. Note however that perturbation experiments by Gudmundsson et

al. 2017 have shown that if the ice melange inside Chasm 1 is removed and replaced by a normal ocean boundary condition, changes to the flow of the ice shelf remain small compared to the 2-fold increase in flow speed since 2012. To avoid confusion, we have reformulated this paragraph.

Fig 1 or 2: As I was reading the results section, I was thinking it'd be nice to clearly visualise the compressive arch. Could you perhaps add a panel (or overlay Fig 1a) showing regions with extension in both directions, versus one compressional component (like Doake et al., 1998).

In the interest of keeping the figures as simple as possible, we had hoped that the reader would be able to approximately trace the outlines of the compressive arch from the arrows in Figure 2, as black arrows are extensive and red arrows are compressive.

Fig 4b: Observations & model match well to the south of the MIR, but the difference grows further north. Can you speculate why?

The key process that causes a slow-down of the ice shelf is the regrounding of the MIR, which is represented well in the model. Further away from the MIR, where the impact of regrounding becomes weaker, other factors come into play, such as temporal changes in the rheology or damage, which are not represented in the model (this has been further clarified in section 6). In particular, further towards the east, just outside the limits of the figure, lies another active rift called the Brunt-StancombWills rift (see e.g. [Gudmundsson et al., 2017]) that locally affects the observed velocities.

**Minor Comments:**
P2L27: A bit pedantic, but I think 'single' would be better than 'singular' here. 'Singular' tends to refer to an exceptional event or thing.

We have removed 'singular' from the text

P3L15: 'preconditions for rifting were re-established'. What were these preconditions? I think the rest of the paper lays out what these preconditions were, but its perhaps a little premature to say this here without explanation.

We added '(as will be explained in section 4)'

P3L18: 'singular' again

We have removed 'singular' from the text

P4L11: slight formatting error – ref in brackets

Brackets have been removed.

Fig 1: North arrow?

As we plotted parallels and meridians (dashed grey lines) in figure 1, we did not see the need to include a north arrow.

Fig 2: Unless is really reduces the clarity of the figures, I'd think that for a colour scale with a white minimum, the minimum ought to be 0 kPa.

In this figure we draw the reader's attention to spatial patterns, with most of the relevant variability between 40 and 130kPa, as reflected in the colour scale. We did not see any benefit in colouring areas with background stresses below 40kPa, as these do not contribute to the story.

---

## Author Comment (AC2) · 28 Aug 2019

**Jeremy Bassis (Referee)** jbassis@umich.edu

**1    Overarching comments**

This study describes a comparison between observations and model inferred stresses for the Brunt ice shelf. The authors demonstrate that the collision of the Brunt ice shelf with a pinning point resulted in increased compressive and tensile stresses and argue that this resulted in the increased tensile stresses needed to reactivate rift propagation. Overall, the results are highly glaciologically relevant and add to our understanding of ice shelf rift propagation.

Thank you for these kind words, and for your in-depth review, which –we believe- has led to important improvements in the presentation of our results.

The manuscript does feel a little bit like it was originally intended for a short form journal

with a restrictive length requirement and has not taken advantage of the more generous space allocated by longer form journals. As such, I had a hard time understanding what was actually done and it seemed as though there were critical details missing from the exposition. Without those details it is hard to assess the reliability of the methods and conclusions drawn. There are also a few places in the manuscript where the text does not appear to accurately reflect the literature or conclusions are not fully supported by the results. My review is relatively long, but most of the issues are (hopefully) easy to correct by expanding the text to include more critical information.

We have expanded the manuscript, in particular the data and methods section, to clarify your points of concern and provide the critical details that were missing. Two appendices were added with additional information about the modelling approach, including details about the inversion method, treatment of the rifts (see below for further comments), and examples of the inferred rate factors.

**2     Major Comments:**

The most significant issue in the manuscript relates to how the authors define a "rift" in the model and how this compares to how the rift actually behaves. We know that rifts in ice shelves can be discontinuities in the ice (fractures) that are filled with ocean water. Rifts are also often filled with a mixture of snow, sea ice and blocks of ice called melange. The melange can become structurally coherent to the point that rifts are barely visible on the surface of the ice, although this tends to be more common in relic rifts that have not been active for decades (or even centuries). Rifts can be represented in models in different ways. The most physically consistent way is to incorporate rifts as actual discontinuities in the ice shelf (see Larour et al., 2014). One then needs to account for the thickness of melange that fills rifts

when applying a normal traction boundary condition along the rift walls, analogous to the calving front boundary condition. Historically, rifts have also been represented in models as regions of intact ice with a rate factor that is set (or inferred) to be much lower than is traditional for intact ice. In this representation, rifts not discontinuities and these features essentially behave like diffuse zones of really warm ice and NOT as fractures. The authors need to be clear about how rifts are represented in their model before any inferences can be made about the effect of "rifts" on the dynamics of ice shelves.

We agree that our *diagnostic* estimates of maximal principal stress, based on snapshots of observed ice velocity, ice thickness and an inferred rate factor (using inverse methods), are not obviously independent of the computational mesh. As you point out, a continuous mesh with soft ice filling the rifts or a mesh with holes might produce a different stress balance. We discuss this issue in Appendix B. As expected, inversions for a continuous mesh produce high values of the rate factor along the trajectory of active rifts (Figure B1, top left panel), whereas for the mesh with holes, these high values are absent (Figure B1, top right panel). However, the derived large-scale stress distribution of the ice shelf remains qualitatively similar between both cases (Figure B1, bottom panels). This is perhaps not surprising as the stresses are strongly controlled by the strain rates, which are forced to be identical for both meshes, by nature of the inverse method.
On the other hand, results from *transient* simulations in section 6 do not include any rift dynamics and we did not address the important question of how active rifts are treated in a transient run – mesh splitting, ice softening, … We have clarified this point in section 6.

A second, and closely related point, arises from circularity in tuning a model to fit observations and then using the fact that the model fits the observations to argue that the model is appropriate.

We are unsure which part of the manuscript you are referring to. If your comment is related to the transient simulations of ice-shelf growth in section 6, and the agreement between our modelled and observed reduction in surface speed after 10 years (Figure 4), then we believe this a genuine projection. The initial model state is 'tuned' to reproduce the observed surface velocities for the given geometry by means of an optimal distribution of the rate factor, but ice shelf geometry, ice thickness and velocities are freely evolved after that. The advance of the ice front, enhanced ice-to-bed contact at the McDonald Ice Rumples and associated slow-down of the flow are non-trivial and obtained without further tuning. A major caveat in our approach is the treatment of the rate factor, which was deliberately kept constant in our transient runs so we could assess the importance of a suitable 'damage model' or 'calving law' by comparing model output to observations. This rationale is now hopefully clear from section 6.

As noted earlier, there are different ways of representing rifts in a model and it is far from clear that the stress field associated with these different representations of rifts are equivalent. In fact, representing rifts as discontinuities will generate stress concentrations near the tip of the rift that will not be present in models that represent rifts as diffuse zones of soft ice.

We hope to have addressed this point above, and refer to Appendix B, in particular figure B1, for more details.

Moreover, when tuning a model to match observations, it is often possible to absorb all uncertainties and errors into the parameters that are being tuned. For example, in traditional damage mechanics, the damage affects the Cauchy stress and thus would also affect the driving stress on the right hand side of the SSA equations. Similarly, errors in density or briny layers of marine ice could also affect the driving stress. One can, of course absorb this error

into inversions for the rate factor, but it is less clear that the inference of the rate factor is *physically* significant. Here, the authors can do more to make their case by describing how rifts are represented in the model and showing actual maps of the rate factor and, if possible, converting those maps to physical variables, like equivalent ice temperature. Fortunately, the Brunt ice shelf itself appears to be well studied and the authors should be able to compare inversion results with the position of known bands of marine ice inferred by King et al.

We acknowledge the fact that the rate factor A, which has been tuned to minimize the misfit between observed and modelled surface velocities, contains information about a variety of physical variables, including ice temperature and fractures. To demonstrate the physical significance of our results, we have provided several maps of $A$ in Appendix A:

1) Figure A2, left panel: results for 1999, prior to rift formation. Away from shear margins and the fractured ice near the grounding line, we expect most of the variability to be related to the internal structure of the ice shelf. Results do indeed distinguish between the bands of colder meteoric ice and surrounding areas of warmer marine ice, as identified by (Thomas 1973 and King et al., 2018).

2) Figure A2, right panel: results for 2016, after the activation of Chasm 1 and the Halloween Crack. Results are for a continuous mesh, and both rifts stand out as bands of weak ice, to accommodate the discontinuity in the flow field across the rifts. The contrast in stiffness (or temperature) between meteoric ice and marine ice is also apparent here.

3) Figure B1, top row: results for 2016, but comparing results for a continuous mesh (left) to results from a mesh with holes (right). In both cases, the distribution of $A$ is broadly similar, except for the absence of soft ice along the rifts in the case of a discontinuous mesh. Some small areas of soft ice remain, likely because the holes in the mesh were based on manual outlines of the rifts from visible satellite imagery, and we were unable to capture the true extent of the active rifts using this method.

We get into similar issues when the authors argue that rift widening causes stress concentrations ahead of the rift. I don't follow the inference if all the authors have done is tune the model to match observations. The authors can, however, make this inference, if they have instead performed a forward model run and the widening of the rift predicted by the model matches observed widening rates and the stress increase near the tip of the rift matches the inferred stress increase. Again, this points to a need for some explanatory text.

Larour, E., et. al., (2014),Representation of sharp rifts and faults mechanics in modeling ice shelf flow dynamics: Application to Brunt/Stancomb-Wills Ice Shelf, Antarctica, J. Geophys. Res. Earth Surf., 119, 1918-1935, doi:10.1002/2014JF003157

We did not perform any forward model simulations with evolving rift dynamics. Instead all stress maps are based upon a diagnostic analysis of an optimal model state (obtained through an inverse method) for successive velocity and geometry snapshots between 2000 and 2017. From these snapshots we infer significant changes in the stress distribution both before and after rift initiation, as discussed in sections 4 and 5.

**3    Detailed comments**

1. The introduction states that the focus of the manuscript is "on the more commonly neglected internal drivers that underlie rift initiation and propagation." It is far from clear to me that this accurately reflects the literature. For example, Fricker et al., 2002, Joughin and MacAyeal, 2005, Larour et al., 2004; Borstad et al., 2012 and 2017 all examine the glaciological stress as the dominant factor driving rift propagation. (There are many, many more citations. These are just a few examples. In fact, almost all of the literature that I can find points towards glaciological stress as the driver.)

We have provided further references to the literature in the introduction, and stressed the fact that all these studies suggest that stresses are an important driver for rift formation and propagation.

"Previous studies have suggested that glaciological stresses are a major control on rift formation and propagation, see e.g. Fricker et al., 2002, Joughin et al., 2005, Larour et al. 2004, Borstad et al., 2012, 2017, and the build-up of internal stresses within an ice shelf can generate energetically favourable conditions for the formation and propagation of rifts that cut through the full depth of the ice column (Rist, 2002). However, a direct link between changing stress conditions prior to calving, and the location and timing of rifts has not been demonstrated so far. This is in part due to the long characteristic time scales over which stresses evolve (typically multiple decades), and the lack of observational data required to calculate the stresses over the duration of a full calving cycle."

More recently (and in the Cryosphere) Arndt (2018) note the role of pinning points in generating rifts in Pine Island glacier, which seems analogous to the study here. In fact, as far as I can tell, with the exception of the Antarctic Peninsula Ice Shelves (that all seem to have effective social media accounts and PR departments), most of the literature for ice shelves does focus on internal stresses. That said, the controls on rifting remain poorly understood, in part because of the long time scale between calving events make the process difficult to directly observe. This observational deficiency, in my opinion, is one of the more significant

motivations and strengths of the present manuscript and it would be useful to re-emphasize this to readers.

We have expanded our motivation for this study in the introduction, and state that "The long-term observational record of the BIS provides unprecedented coverage of glaciological changes over a full calving cycle, from the last calving event in the early 1970s to present day.". We have stressed the importance of the McDonnald Ice Rumples (local pinning point) for the dynamics and its role as a trigger for calving. We have pointed out why lessons learnt for the Brunt Ice Shelf are lessons learnt for ice shelves elsewhere in Antarctica, and have referenced Arndt (2018) as an example. We will address this point in more detail in our reply to your next comment.

Fricker, HA, Young NW, Allison I, Coleman R. 2002. Iceberg calving from the Amery Ice Shelf, East Antarctica. Annals of Glaciology, Vol 34, 2002. 34

Joughin, I., and MacAyeal, D.R. (2005), Âa Calving of large tabular icebergs from iceˇ shelf rift systems, Âa Geophys. Res. Lett., 32, L02501, doi:10.1029/2004GL020978.ˇ

Larour, E., Rignot, E., and Aubry, D. (2004), Modelling of rift propagation on Ronne Ice Shelf, Antarctica, and sensitivity to climate change, Geophys. Res. Lett., 31, L16404, doi:10.1029/2004GL020077.

Borstad, C., McGrath, D., and Pope, A. (2017), Fracture propagation and stability of ice shelves governed by ice shelf heterogeneity, Geophys. Res. Lett., 44, 4186–4194, doi:10.1002/2017GL072648.

Borstad, C.P. et al. A damage mechanics assessment of the Larsen B ice shelf prior to collapse: Toward a physically-based calving law. Geophy. Res. Lett. 39, L18502 (2012).

Arndt, J. E., Larter, R. D., Friedl, P. , Gohl, K., Hoppner, K.Âaand¡ athe Science Teamˇ of Expedition PS104, (2018): Bathymetric controls on calving processes at Pine Island Glacier , The Cryosphere, 12(6), pp. 2039–2050. doi:10.5194/tc-12-2039-2018

2. Iceberg calving is a natural process and this needs to be more clearly emphasized

The broader context in the introduction is the looming calving event from the Brunt Ice Shelf. Building on the previous point, it is known that ice shelves exhibit a natural cycle of decades to centuries advance punctuated by episodic retreat associated with the detachment of large tabular icebergs (see, Fricker et al., 2002, Walker et al., 2015). These calving events are believed to be driven by a combination of the accumulation of fractures coupled with changes in the glaciological stress. Yes, there is evidence for climate driven disintegration of ice shelves, primarily on the Antarctic Peninsula, but this is more of an exception to the norm. I suggest that the authors consider adding more context to the introduction, explaining not only that calving is part of the natural cycle of ice shelves, but how the calving event from the Brunt Ice Shelf fits into this larger context. How similar is this event from previous events? Or are are there no records of previous events? How does the cycle compare to other ice shelves?

The second paragraph of the introduction now starts with "Calving events are part of the natural life cycle of all ice shelves, as they go through internally-driven periods of growth and collapse (see e.g. Fricker et al., 2002, Anderson et al., 2014, Hogg et al., 2017)." and we have provided a broader context for the calving of the Brunt: "….As such, the cyclic dynamics of the BIS is modulated by natural changes in ice shelf geometry, and observations indicate that each cycle lasts approximately 40-50 years, which is comparable to other stable ice shelves (see e.g. Fricker et al., 2002).

The significance of local grounding at the MIR for the dynamics of the BIS, and its role in recent rifting events will be explored in subsequent chapters. However, the wider importance of pinning points for the dynamics and structural integrity of Antarctic ice shelves has been previously recognised (Borstad et al., 2013, Matsuoka et al., 2015, Favier et al., 2016, Berger et al., 2016, Gudmundsson et al., 2017), and their potential role in triggering calving events was highlighted recently for Pine Island Glacier (Arndt et al, 2018). Here we use the BIS as an example to demonstrate the link between naturally evolving glaciological conditions, the initiation of ice-shelf rifts, and the mechanical drivers that govern subsequent rift propagation. The geometrical configuration of the BIS is not unique, and similar principles likely apply to other Antarctic ice shelves that are dynamically constrained by local pinning points, such as the ice shelves in Dronning Maud Land (Favier et al., 2016) and the Larsen C Ice Shelf (Borstad et al., 2013). Our study is more generally relevant for ice shelves that experience a build-up of stress, potentially far upstream of the ice front, due to natural changes in ice-shelf geometry."

Similarly (and sorry for being pedantic), one of the issues that hindered my understanding of the broader context of the study is that the term "unique" is used frequently (4 times at least) and it was unclear what, exactly was unique in each of these instances?

The first time unique was used, it was used to describe the 50-year time series. This seems like appropriate usage. But, the next time we are told there is "a unique opportunity to enhance . . . process-based understanding". What is unique about the opportunity? Is this the 50-year time series? If this calving event is a continuation of the natural cycle, then (pedantically), the opportunity is not unique. There are also other rifts

on other ice shelves that have been (or can be) studied. What exactly is unique about this opportunity/rift? The third time we are told there is a "unique, network of up to 15 GPS". GPS have been deployed around rifts (propagating and not propagating) in ice shelves multiple times so what about this particular deployment is unique. Finally, we are told "BIS represents a unique setting . . . calving processes can be studied" This sounds like the authors are arguing that rift propagation/iceberg calving is different in this situation than the calving cycle that is observed elsewhere? It would be helpful to clarify all of this.

We understand your confusion and have either replaced "unique" by more appropriate wording, or removed it altogether. For example, we have replaced "Fortuitously, a **unique** opportunity to enhance our process-based understanding of rift dynamics and calving has recently arisen" by "Fortuitously, a **new** opportunity to enhance our process-based understanding of rift dynamics and calving has recently arisen"

Overall, I think that the authors could help readers understand the significant of the Brunt Ice Shelf and the particular rift system by sketching out what is common about the iceberg calving process across ice shelves. Then, tell us what is unusual in this situation (is it just the observations? the pinning point?) and what is truly unique here (is Brunt itself unique due to the large heterogeneities?). This would enable readers to better understand how this study fits into the broader context of rifting and calving from other ice shelves.

Walker, C., Bassis, J., Fricker, H., Czerwinski, R. (2015). Observations of interannual and spatial variability in rift propagation in the Amery Ice Shelf, Antarctica, 2002–14. Journal of Glaciology, 61(226), 243-252. doi:10.3189/2015JoG14J151.

We hope to have addressed this point in the introduction, as quoted above.

**3. Methods, part 1 (inversions)**

This is the where I really started to struggle to understand what was done and there is critical detail missing from the description of the model and inversion process. The model is described as a shallow-shelf approximation model SSA, which is standard for ice shelves. My understanding of the inversion is that the authors invert for the rate factor A(x,y) by ingesting surface velocities into the model. However, the inversion uses two regularization parameters $\gamma_a$ and $\gamma_s$ neither of which are defined in the text. Digging into Reese et al., 2018, it looks like the regularization parameters correspond to the ice softness AND basal friction coefficient. But an ice shelf, by definition, is freely floating and there is no basal friction. Are the authors inverting for basal friction beneath the pinning points? Is this done everywhere or in certain places? More details and more clarity are needed to understand what has been done here. I now see all the way at Page 8 that a Weertman sliding law is used specifically for the pinning points. This needs to be explained much earlier if it fits into the inversions. Also, what is the shape of the pinning point? Is the ice shelf plowing over it or is the pinning point just tickling the bottom?

We have rewritten this part of the methods and data section, and added a dedicated Appendix A with more details about the inverse method. The regularization parameters were defined, and results from an L-curve approach were added to motivate the chosen values. The description of the sliding law was moved to section 3, and additional details were added about the bedrock topography underneath the pinning point.

"Due to the inaccessibility and complex topography of the surface at the MIR, ground-based and airborne radar surveys have failed to reliably measure the bedrock topography in this location (Hodgson et al., 2019). In our analysis, the elevation of the bed was therefore set to 10m above the floatation depth across the extent of the MIR, and basal traction between the bed and ice was parameterized by a Weertman sliding law. The latter provides a commonly

adopted relation between the basal sliding velocity vb and basal shear stress τb in grounded areas, $\tau_b = C^{(-1/m)} v_b^{(1/m-1)} v_b$, with m and C model parameters. A common value for the sliding exponent m = 3 was chosen, and the slipperiness coefficient was set to a spatially uniform value C = 10−3."

4. Deviatoric stress is not constant with depth

There are also more subtle issues associated with the interpretation of the inferred rate factor. In the SSA approximation strain rate is independent of depth. However, the stress is only independent of depth if the temperature in the ice is constant within each column of ice. What the authors are really inferring is the depth averaged rate factor. A consequence is that the authors are also only able to show the depth averaged deviatoric stress. Stress could be much higher near the surface of the ice, where temperatures are likely much colder. This needs to be recognized and explained and in particular, related back to the physical interpretation of the rate factor of ice.

We acknowledge this fact and have made clear in the paper that *A* represents a vertically averaged value: "In general, ***A varies spatially over several orders of magnitude (both horizontally and vertically),*** and an alternative approach for estimating τ relies on commonly-used inverse theory, which uses observations of ice shelf geometry, velocity and ice thickness data to estimate an optimal spatial distribution of the rate factor *A(x)* by minimizing the mismatch between observed and simulated ice velocities (see e.g. MacAyeal, 1993 and Larour et al., 2005). The resulting solution for *A* and the diagnostic model output for $\dot\epsilon$ can be used to calculate a spatial distribution of the deviatoric stress τ and its principal components**. We used an adjoint iterative optimization method with Tikhonov regularization within the SSA (Shallow Shelf Approximation) ice flow model Úa (Gudmundsson et al, 2012) to obtain vertically-integrated values for *A(x)* and *τ(x)*, where *x* denotes both horizontal dimensions**. Further details about the model setup, the inversion procedure and examples of *A(x)* for various ice-shelf configurations can be found in Appendix A."

5. Methods, part 2 (maps of rate factor please)

It helpful to readers to see the actual maps of inferred ice softness and basal friction (if this was also inverted for). This would certainly help convince readers that patterns of rate factor are realistic and not spurious artifacts. This is a matter of preference, but I personally also like to see the inferred rate factor converted into an ice temperature so that we can be sure that the ice temperature is semi-realistic based on known con ditions. The authors note that these are related to structural properties of the ice. In particular, it would be helpful to know if the inversions for ice softness correspond to regions of marine ice documented by King et al., (2018). In fact, one also wonders if the inversion could resolve the sharp variations in material properties associated with the bands of marine ice documented by King et al., (2018). A standard way to test this is by doing a "checkerboard" test. You compute the forward model using a checkerboard or other pattern. You then add noise to the signal and invert based on the synthetic data. This would give a sense of the resolution of the inversion and if the inversion can pick up relevant structural features. The more formal way of doing this would be to construct resolution kernels to formally determine what can and cannot be resolved.

We have provided maps of *A* and corresponding 'ice temperatures' for several ice-shelf configurations, both before and after rift initiation, in Appendix A. We comment on the physical meaning of these values, and discuss the broad spatial patterns, which indeed reflect previously identified variations in ice shelf structure (King et al., 2018).

6. Observations, how do you shift the data to a date?

I thought the observation section was much clearer and easier to understand. But it was unclear to me how you "shift" a DEM to an effective time step? There is no reference or description of the method used to do this. This, along with any error associated with the procedure should be described.

This has now been described in more detail in section 3:

"To correct for the ice motion between the different acquisition times of the WorldView-2 tiles, all tiles were translated to a common datum of 1 January 2013. For each tile, pixels were shifted by $\Delta x = u\Delta t$ with $u(x)$ the surface velocity at location x obtained from a pair of Sentinel-1 images acquired in June 2015, and $\Delta t$ the difference between the acquisition time of the tile and the common datum."

7. Can a viscous ice shelf model really accumulate stress at the tip of a rift by dissipating gravitational potential energy?

We did not include any prescription of dynamical rift evolution in our transient runs in section 6, but instead kept the rate factor constant and focussed on the discrepancies between observations and model output that resulted from the lack of an appropriate description of rift initiation and evolution. All maps of deviatoric stress in sections 4 and 5 are snapshots, based on an optimal estimate of the rate factor for given surface velocities and ice shelf geometry.

The authors argue that rift widening results in accumulation of stress ahead of the rift. The energy balance in an ice shelf model tells us that gravitational potential energy is dissipated through viscous flow. The accumulation of stress seems to imply energy is being added to the system faster than it can be removed. What is the source of the energy that is added to the system that drives energy accumulation? Is this related to torques associated with rotation of the blocks of ice? Is this conclusion supported by forward model runs or this based on tuning the model to match observations? Given the fact that rift widening is documented by GPS, it seems as though the authors should be able to do a forward model run to compare simulations with observed rift widening and use the forward model to show that stress is concentrated ahead of the rift.

Both Chasm 1 and the Halloween Crack have been observed to widen at slowly accelerating rates for weeks to months, without noticeable propagation (De Rydt et al., 2018). The widening of both rifts results from the rotational motion of the soon-to-be-icebergs away

from the main ice shelf, and the resulting torque is either dissipated through viscous deformation or stored until further rift propagation occurs. We did not estimate the energy balance or perform a forward simulation with a description of the rift dynamics, which would generally require a viscoelastic treatment with fracture propagation criteria such as VCCT or the J-integral. Instead our (much less ambitious) aim was to detect changes in the far-field stress distribution of the ice shelf through snapshot inversions of surface velocity and ice shelf geometry using an SSA model, and diagnose the optimal model state for its stress configuration at various times. The results are presented in sections 4 and 5.

8. Conclusions

This is up to the author, but there has been little doubt that "calving laws" are needed in ice sheet models and this is not the main conclusion I would draw from this study. I don't think it will come as any surprise to most readers that iceberg calving is an important process in ice sheet evolution. The authors also have a typo in their description of the so-called marine ice cliff instability. The marine ice cliff instability assumes that there is a **maximum** ice thickness, not a **minimum** ice thickness. Moreover, the marine ice cliff instability generally applies to thick grounded ice and not thin ice shelves, like the Brunt Ice Shelf. Minimum ice thickness models have been a mainstay in ice sheet models for decades largely as a means of preventing ice shelves from indefinitely advancing. Hence, the authors have a good argument that these minimum thickness criteria are not physical.

Actually, coming back to the Deconto and Pollard marine ice cliff instability study, my understanding is that the parameters used by Deconto and Pollard were derived based on parameter sensitivity studies for past sea level rise. These are, technically, observations are they not? Direct observations of ice flow of ice sheets hundreds of thousands of years ago, similar to the GPS and satellite imagery used in this study, remains a challenging problem in paleo ice sheet studies. And if the marine ice cliff instability is really a thing, the only evidence we have likely comes from past ice sheet conditions when these processes may have been active. Here, it would be useful if the authors put their results in context of past and future projections of ice sheet change. If structural heterogeneity is important, is it possible to predict it instead of tuning a model to match observations? How important is structural heterogeneity versus the geometry of pinning points? It seems like knowing the location of pinning points (which is possible) could at least provide a first order approach to rift generation even if it does not match the detailed sub-decadal trends? This study potentially offers a lot of information and it would be useful to readers to see how this fits into the bigger picture.

We think these are all very valid points and have reformulated the conclusions to better reflect these thoughts:

"Our results, based on observations and numerical modelling, demonstrate how ice shelves that are dynamically constrained by local pinning points, such as the Brunt Ice Shelf, can experience significant changes in internal stress caused by their naturally evolving geometry, and generate favourable conditions for rifting far upstream of the ice front. Such conditions make these ice shelves particularly susceptible to fast collapse, a process that is not generally captured by present-day ice flow models, despite recent progress (Levermann et al., 2012; Borstad et al., 2012). Existing calving criteria based on a maximum ice thickness, such as the marine ice-cliff instability mechanism (De Conto and Pollard, 2016), remain controversial and might not be directly relevant for thin floating areas such as the BIS. Other commonly-used calving laws based on minimum ice thickness criteria discard variations in mechanical properties of the ice, and are independent of internal stress. Existing theories for the vertical propagation of surface and basal crevasses (Hughes, 1983, van der Veen, 1998a, 1998b), often linked to surface hydrology (Scambos et al, 2000, Scambos et al., 2009, Nick et al., 2013), do not generally include criteria for the initiation and horizontal propagation of full-depth rifts. As a result, model simulations do not generally capture rapid and large-scale changes in ice shelf geometry, and thereby underestimate the critical role of ice shelves as a buffer against further mass loss from the Antarctic Ice Sheet."

Minor comments:

Page 5, line 25: What is "Geometric Deformation"? Do the authors mean that the geometry of the ice shelf is changing? I actually googled this term, but all of the hits directed me to papers on differential geometry, which seems like it is not what the authors are talking about.

Changed to "changes in the ice shelf's geometry".

Page 6, line 26. Ice is in hydrostatic equilibrium. A force balance at the ice-ocean interface (analogous to that at the calving front) within a melange free rift suggests a deviatoric stress pointing into the rift. Why is the ocean pressure pulling the rift apart? The large scale stress of the ice shelf might pull the ice apart. This should be clarified.

We have replaced this line by a more precise statement "Following rift propagation, newly formed rift surfaces were subjected to ocean pressure, and stresses within the ice shelf gradually adjusted to the new boundary conditions and newly emerging ice front location. In particular, maximum tensile stresses aligned perpendicular to the edges of the rifts."

Page 3, line 28: We were told there is 50 years of data, why only focus on the period from 1997-2018? What is the benefit of the long time series if less than half are used? The earlier emphasis on 50 years of data seems like a bit misleading at this point.

Different parts of the long-term record have been used for different purposes. The 1915-1970s data have provided valuable context for present-day events because it has allowed us to describe the ongoing calving as 'reoccurring'. The 1999-2017 data with full spatial coverage of the ice shelf, on the other hand, has allowed us to analyse ice-shelf wide changes in the flow and stresses before and after rift initiation. We have changed the abstract to " …20-years' time series of in-situ and remote sensing observations.." but have put emphasis on the use of the long-term record at various places in the introduction.

Page 1, line 20: This is pedantic, but I would consider the 5000 km$^2$ berg that detached from the Larsen Ice Shelf to be a small to mid-sized berg. Iceberg B15 that detached from the Ross Ice Shelf was twice as large and Shackleton documented icebergs that were even larger.

We meant 'large' compared to the size of calving events caused by bending stresses near the ice front.

Page 1, line 20: The word "since" refers to time. For example, "It has been a long time since the Knicks won the championship." In this case, I believe you want "Because".

Corrected.

Page 2, line 9: The references given here document thinning of ice shelves and do not appear to describe any links between thinning and calving.

This paragraph has been removed.

Everywhere: space between numbers and units 3m should be 3 m

Corrected.

Page 7, line 2: comma after "but"

Corrected.

Page 7, 2nd paragraph: Now I'm really confused about what is going on. Are the authors introducing a rift into the model and widening it, based on observations to examine the stress field. Or, have the authors inverted for stress (OK, actually ice softness) based on surface velocities at several intervals of time? In the first case, I think the authors are safe saying that the increase in stress is due to rift widening. In

the second case, I don't know that you can say that the stress is caused by widening when no rift widening has been included in the model and the model has been tuned to reproduce surface velocities (and hence stresses). Morever, the assumption that rift widening results in stress concentration should be checked against other periods of time when rift propagation did not occur. For example, there is a long history of rift widening without propagating prior to Chasm's reactivation. Does this period of time correspond to the rift propagating into a zone of marine ice?

We follow the second approach, i.e. invert for stresses to match different snapshots of surface velocity and ice shelf geometry (incl. rift extent, ice front location and surface elevation). Based on this method, significant changes in stress are found between successive time stamps. These are interpreted as the combined effect of changes in local grounding at the MIR, changes in rift extent and width, and overall deformation of the ice. We agree that they cannot be attributed to rift widening alone, and we have removed this statement from the paper.

Page 7, line 26: The technical jargon "damage" is introduced here. Authors should avoid the term or define it. Keep in mind that "damage" has a precise definition in the fracture mechanics literature and is, most generally, a tensor. The term damage is often used heuristically in glaciology in confusing and imprecise ways. If the authors mean rifting, I recommend just saying rifting.

We have replaced the term "damage" by "rift" or "fracture" throughout the manuscript.

Section 6, Page 8, section paragraph: Wait a minute. Why is the rate factor A(x,y) not a property of ice that advects with the ice? Conventionally, the rate factor has been linked to temperature, grain size and crystal structure of the ice. If reductions in the factor A(x,y) are linked to fractures then surely these must also advect with the ice? If the advection of the rate factor is not important, then why is heterogeneity of the ice important? I'm missing something critical here because this seems like this contradicts the authors main conclusion that heterogeneity is important.

Rather than simulating changes in $A$ to reflect fractures and aiming to reproduce observations as closely as possible, in section 6 we analysed discrepancies between observations and transient model output for a constant rate factor. As we pointed out in the paper, projections with constant $A$ over

century to millennium timescales are still common practice, and this approximation might not be justified in certain cases. Our experiment in section 6 led to two main results: 1) between 2000 and 2010 (i.e. before rift initiation) the model is capable of reproducing the observed slow-down of the Brunt Ice Shelf as a result of increased buttressing at the pinning point. In other words, the SSA model does an ok job at reproducing the conditions that eventually caused fracture formation, even with a constant rate factor.  2) After 2012 and the initiation of Chasm 1 and the Halloween Crack, model projections with a constant rate factor started to deviate significantly from observations (up to 100%). This was expected, and almost entirely due to the lack of a suitable description of fracture initiation and propagation. Our approach has allowed us to quantify errors related to the lack of an appropriate treatment of fractures (by means of changes to *A* or other), and has demonstrated the need for such a treatment for certain ice shelves, even at decadal timescales.

The "extrusion" method for calving front advance is known to generate significant artifacts if not treated carefully. The calving front should advect as a sharp shock and accurate shock capturing methods are needed to avoid overly diffusing the calving front. Numerical details of advection should be included with limitations described. Does the advection scheme preserve mass? It is diffusive? Does it preserve the shock-like nature of the calving front? Are results sensitive to grid size or time step size?

In the Figure below we show sections of the transient ice shelf geometry for two flowlines: at the top is a flowline towards the east of the pinning point (Gaussian bump in the figure), and at the bottom is a flowline through the middle of the pinning point. Different colours indicate different years, with blue curves corresponding to the (ungrounded) 2000 geometry, and red the 2010 geometry after re-grounding onto the pinning point. As can be seen from the top panel, the ice front remains relatively well-defined when not distorted by the bedrock, as is the case in the lower panel. As pointed out in the paper, "A fully implicit time integration with streamline upwind Petrov-Galerkin method and stabilization (SUPG) was used, and the ice front was found to advance with limited diffusion or spurious oscillations."

[Figure]

Page 8, last paragraph: The statement that ice sheet models keep the calving front pinned to present day conditions might have been true a decade or two ago, but pretty

much all of the major ice sheet models at this point allow the calving front to evolve. PISM uses a wetting drying algorithm combined with "eigen calving". ISSM uses a level set method combined with a Von Mises calving law. BISICLES and CISM have their own methods to advance the calving front and use a spectrum of calving laws. These days, models allow the calving front to advance and retreat according to heuristic (and often known to be incorrect) parameterizations. Whether advancing and retreating the calving front based on inaccurate and unphysical calving laws is progress is a question that I will leave to others.

We agree that several models have at least some capability to advance or retreat their ice fronts. However, these routines are not regularly used with confidence or not used at all, hence our statement. To pick up on this point, we have reformulated our statement as follows: ""

Page 12, line 12: Reference to Lipovsky, 2018b appears to reference an unpublished manuscript. Check Cryosphere style guidelines for rules on references to non-peer reviewed literature. This is prohibited by AGU publications, but the standards of TCD might not be as stringent

Reference has been removed.

Figure 2-3. Best not to use a red-green color scale.

This color scale was tested by and deemed suitable for colorblind people, unless there are other reasons not to use a red-green color scale?